# AGENTCLINIC: A MULTIMODAL AGENT BENCHMARK TO EVALUATE AI IN SIMULATED CLINICAL ENVIRONMENTS

## ABSTRACT

Evaluating large language models (LLM) in clinical scenarios is crucial to assessing their potential clinical utility. Existing benchmarks rely heavily on static question-answering, which does not accurately depict the complex, sequential nature of clinical decision-making. Here, we introduce AgentClinic, a multimodal agent benchmark for evaluating LLMs in simulated clinical environments that include patient interactions, multimodal data collection under incomplete information, and the usage of various tools, resulting in an in-depth evaluation across nine medical specialties and seven languages. We find that solving MedQA problems in the sequential decision-making format of AgentClinic is considerably more challenging, resulting in diagnostic accuracies that can drop to below a tenth of the original accuracy. Overall, we observe that agents sourced from Claude-3.5 outperform other LLM backbones in most settings. Nevertheless, we see stark differences in the LLMs' ability to make use of tools, such as experiential learning, adaptive retrieval, and reflection cycles. Strikingly, Llama-3 shows up to 92% relative improvements with the notebook tool that allows for writing and editing notes that persist across cases. To further scrutinize our clinical simulations, we leverage real-world electronic health records, perform a clinical reader study, perturb agents with biases, and explore novel patient-centric metrics that this interactive environment firstly enables.

## 1 INTRODUCTION

One of the primary goals in Artificial Intelligence (AI) is to build interactive systems that are able to solve a wide variety of problems. The field of medical AI inherits this aim, with the hope of making AI systems that are able to solve problems which can improve patient outcomes. Recently, many general-purpose large language models (LLMs) have demonstrated the ability to solve hard problems, some of which are considered challenging even for humans (Thirunavukarasu et al., 2023). Among these, LLMs have quickly surpassed the average human score on the United States Medical Licensing Exam (USMLE) in a short amount of time, from 38.1% in September 2021 (Gu et al., 2021) to 90.2% in November 2023 (Nori et al., 2023) (human passing score is 60%, human expert score is 87% (Liévin et al., 2023)). While these LLMs are not designed to replace medical practitioners, they could be beneficial for improving healthcare *accessibility* and scale for the over 40% of the global population facing limited healthcare access (Organization et al., 2016) and an increasingly strained global healthcare system (McIntyre & Chow, 2020).

However, there still remain limitations to these systems that prevent their application in real-world clinical environments. Recently, LLMs have shown the ability to encode clinical knowledge (Singhal et al., 2023; Vaid et al., 2023), retrieve relevant medical texts (Xiong et al., 2024), and perform accurate single-turn medical question-answering (Liévin et al., 2022; Nori et al., 2023; Wu et al., 2023; Chen et al., 2023). However, clinical work is a multiplexed task that involves sequential *decision making*, requiring the doctor to handle uncertainty with limited information and finite resources while compassionately taking care of patients and obtaining relevant information from them. This capability is not currently reflected in the static multiple choice evaluations (that dominate the recent literature) where all the necessary information is presented in a case vignettes and where the LLM is tasked to answer a question, or to just select the most plausible answer choice for a given question.

In this work, we introduce AgentClinic, an open-source multimodal agent benchmark for simulating clinical environments. We improve upon prior work by simulating many parts of the clinical environment using language agents in addition to patient and doctor agents. Through the interaction with a measurement agent, doctor agents can perform simulated medical exams (e.g. temperature, blood pressure, EKG) and order medical image readings (e.g. MRI, X-ray) through dialogue. We also support the ability for agents to exhibit 24 different biases that are known to be present in clinical environments. We also present environments from 9 medical specialties, 7 different languages, and a study on incorporating various agent tools and reasoning techniques. Furthermore, our evaluation metrics go beyond diagnostic accuracy by giving emphasis to the patient agents with measures like patient compliance and consultation ratings.

Our key contributions are summarized as follows:

1. We challenge how large language and vision models should be evaluated for medical diagnosis with the introduction of AgentClinic. These diagnostic challenges are not static QAs, but are interactive, dialogue-driven, sequential decision making environments that require data collection, ordering appropriate medical exams, and understanding medical images across patients with unique family histories, lifestyle habits, age categories, and diseases.

2. A system for incorporating complex **biases** that can affect the dialogue and decisions of patient and doctor agents. We present results on diagnostic accuracy and patient perception for agents that are affected by cognitive and implicit biases. We find that doctor and patient biases can lower diagnostic accuracy, affect the patient's willingness to follow through with treatment (compliance), reduce patient's confidence in their doctor, and lower willingness for follow-up consultations.

3. We introduce patient agents built from real clinical cases sourced from electronic health record data, including an agent-based system for providing simulated medical exams (e.g. temperature, blood pressure, EKG) based on realistic disease test findings. We also introduce patient cases from nine **medical specialties** and seven **multilingual** environments to better support specialist applications and diverse language backgrounds. We also present realism and empathy ratings from clinicians for the resulting dialogue.

4. We allow doctor agents to use a variety of **tools**, such as browsing the web, textbooks, perform reflection cycles, take and edit notes in a notebook that persists over different patient scenarios. We show that current LLMs vastly differ in how much they benefit from these tools, with some models demonstrating large accuracy increases while others decrease in accuracy.

## 2 AGENTCLINIC: A MULTIMODAL AGENT BENCHMARK FOR CLINICAL DECISION MAKING

In this section we describe AgentClinic, which uses LLM agents to simulate a clinical environment.

**Language agents** Four language agents are used in the AgentClinic benchmark: a patient agent, doctor agent, measurement agent, and a moderator (Figure 1). Each language agent has specific instructions and is provided unique information that is only available to that particular agent. These instructions are provided to an LLM which carries out their particular role. The doctor agent serves as the model whose performance is being evaluated, and the other three agents serve to provide this evaluation. A detailed description of each agent is provided in Appendix A.1.

**Language agent biases** Previous work has indicated that LLMs can display racial biases (Omiye et al., 2023) and might also lead to incorrect diagnoses due to inaccurate patient feedback (Ziaei & Schmidgall, 2023). Additionally, it has been found that the presence of prompts which induce cognitive biases can decrease the diagnostic accuracy of LLMs by as much as 26% (Schmidgall et al., 2024). The biases presented in this work intend to mimic cognitive biases that affect medical practitioners in clinical settings. However, these biases were quite simple, presenting a cognitive bias snippet at the beginning of each question (e.g. "*Recently, there was a patient with similar symptoms*

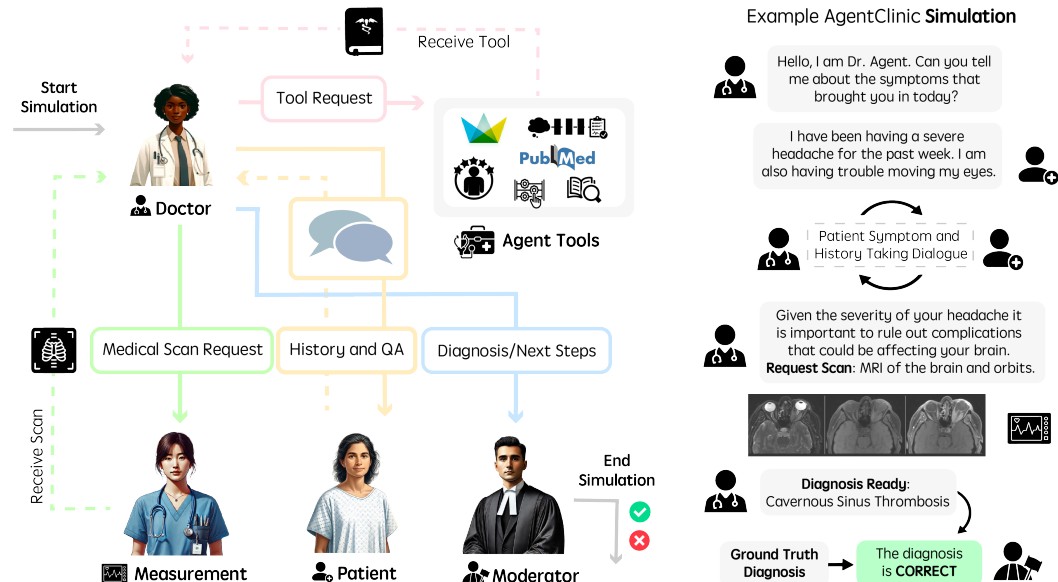

Figure 1: Running language agents in AgentClinic. (Left) Workflow diagram of agents in AgentClinic. The doctor agent interacts with tools and agents in order to arrive at a diagnosis. Moderator agent compares conclusion to ground truth diagnosis at the end of the simulation. (Right) Example dialogue between agents in the AgentClinic benchmark.

*that you diagnosed with permanent loss of smell*"). This form of presentation did not allow for the bias to present in a realistic way, which is typically subtle and through interaction.

We present clinically relevant biases that have been studied in other works from two categories: cognitive and implicit biases (Fig. 7). Cognitive biases are systematic patterns of deviation from rational judgment, such as recency bias, where recent cases disproportionately influence clinical decisions, or anchoring bias, where early diagnostic impressions overly dictate later assessments. Implicit biases, on the other hand, are unconscious associations shaped by societal and cultural norms. These include biases based on race, gender, or socioeconomic status, which can subtly influence the quality of patient interactions and treatment plans. These biases are introduced by adding context into the agent's system prompt instructing them to play out that bias as part of their role. For instance, to simulate sexual orientation bias, the patient agent receives the prompt: "*You are uncomfortable with your doctor because you find out that they are a particular sexual orientation and you do not trust their judgement. This affects how you interact with them.*" This is discussed in Appendix A.2.

**Building agents for AgentClinic** In order to build agents that are grounded in medically relevant situations, we use a random sample of diagnostic questions from the US Medical Licensing Exam (USMLE), from deidentified electronic health records (MIMIC-IV) (Johnson et al. (2023)), and from the New England Journal of Medicine (NEJM) case challenges. These questions are concerned with diagnosing a patient based on a list of symptoms, which we use in order to build the Objective Structured Clinical Examination (OSCE) template that our agents are prompted with. For AgentClinic-MedQA and AgentClinic-MIMIC-IV, we first select from a sample of questions from the MedQA and MIMIC-IV dataset respectively and then populate a structured JSON formatted file containing information about the case study (e.g. test results, patient history) which is used as input to each of the agents. The exact structure of this file is demonstrated in Appendix I as well as an example case study shown in Appendix J. In general, we separate information by what is provided to each agent, including the objective for the doctor, patient history and symptoms for the patient, physical examination findings for the measurement, and the correct diagnosis for the moderator. We initially use an LLM (GPT-4) to populate the structured JSON, and then manually validate each of the case scenarios. For AgentClinic-NEJM we select a curated sample of 120 questions from NEJM case challenges and proceed with the same template formatting as AgentClinic-MedQA/MIMIC-IV.

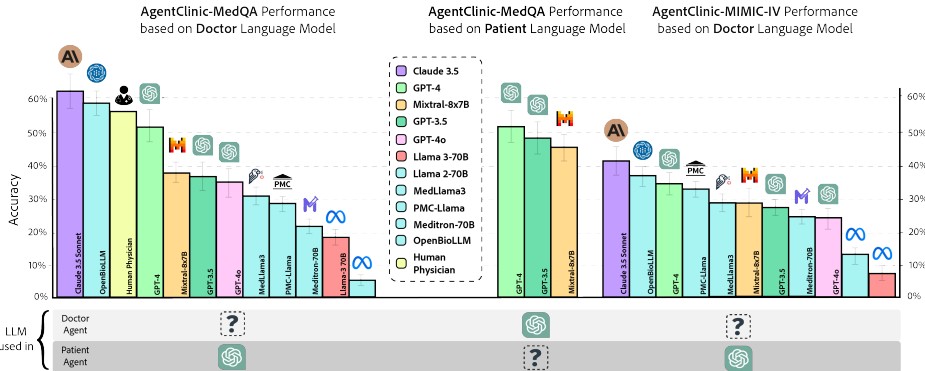

Figure 2: Accuracy of various **doctor** language models and human physicians on AgentClinic-MedQA using GPT-4 patient and measurement agents (left). Accuracy of GPT-4 on AgentClinic-MedQA based on **patient** language model (middle). Accuracy on AgentClinic-MIMIC-IV by number of using GPT-4 patient and measurement agents (right).

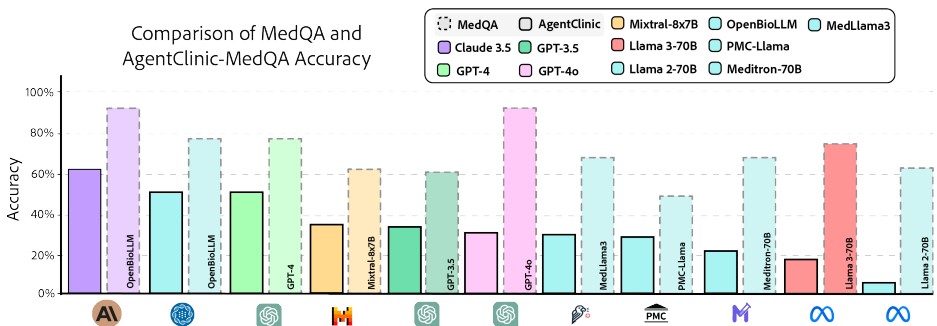

Figure 3: Comparison of accuracy of models on MedQA and AgentClinic-MedQA. We find that MedQA accuracy is not predictive of accuracy on AgentClinic-MedQA.

**Multilingual and Specialist cases**   Multilingual patient cases are converted from AgentClinic-MedQA to the the target language using GPT-4 and then manually corrected by native speakers. Agents are then prompted to perform dialogue in the target language. We chose to focus on six languages: Chinese, Hindi, Korean, Spanish, French, and Persian. The selection of these languages aims to address the need for medical AI systems capable of operating in multilingual healthcare environments. Specialist cases use case report questions from the MedMCQA dataset (Pal et al. (2022)). These questions include case reports from 20 different medical specialties, from which we chose to focus on 9 patient-focused specialties in AgentClinic-Spec: emergency medicine, geriatrics, pharmacology, internal medicine, psychiatry, ophthalmology, otolaryngology, and pediatrics.

## 3   RESULTS

### 3.1   COMPARISON OF MODELS

Here we discuss the accuracy of various language models on AgentClinic-MedQA. We evaluate 11 models in total: Claude-3.5-Sonnet, GPT-4, GPT-4o, Mixtral-8x7B, GPT-3.5, Llama 3 70B-Instruct, Llama 2 70B-chat, MedLlama3-8B, PMC-Llama-7B, Meditron-70B, and OpenBioLLM-70B (model details discussed in Appendix C). Each model acts as the doctor agent, attempting to diagnose the patient agent through dialogue. The doctor agent is allowed N=20 patient and measurement interactions before a diagnosis must be made. We also evaluate human physician performance collected from three physicians, provided the same instructions and constraints as the LLMs. For this evaluation, we use GPT-4 as the patient agent for consistency. The accuracy of each models is presented in Figure 2: Claude-3.5 62.1% $\pm$ 3.3, OpenBioLLM-70B 58.3 $\pm$ 4.2, Human Physicians 54

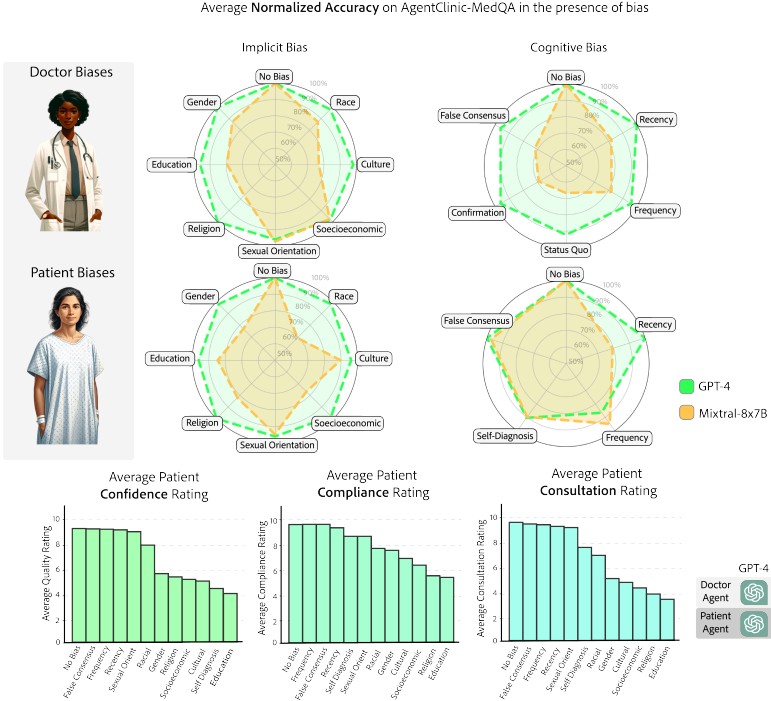

Figure 4: **(Top)** Demonstration of normalized accuracy (Accuracy$_{\text{bias}}$ / Accuracy$_{\text{No Bias}}$) with implicit and cognitive biases with GPT-4 (green) and Mixtral-8x7B (orange). GPT-4 accuracy was not susceptible to biases, whereas Mixtral-8x7B was. **(Bottom)** Ratings provided after diagnosis from GPT-4 patient agents with presented biases. *Left.* Patient confidence in doctor. *Middle.* Patient compliance, indicating self-reported willingness to follow up with therapy. *Right.* Patient consultation rating, indicating willingness to consult with this doctor again.

$\pm$ 28.5, GPT-4 at 51.6% $\pm$ 3.3, Mixtral-8x7B at 37.1% $\pm$ 3.1, GPT-3.5 at 36.6%, GPT-4o 34.2% $\pm$ 3.4, MedLlama3-8B 31.4 $\pm$ 2.9, PMC-Llama 7B 23.6 $\pm$ 2.1, Meditron 70B 29.1 $\pm$ 2.4, MedLlama3-8B 31.4 $\pm$ 2.9, Llama 3 70B at 19% $\pm$ 2.5, and Llama 2 at 70B-chat 4.5% $\pm$ 1.3. Confidence intervals for all experiments are provided in Appendix D.

We use the same configuration for AgentClinic-MIMIC-IV, with model accuracy presented in Figure 2: Claude-3.5 42.9% $\pm$ 3.3, GPT-4 34.0% $\pm$ 3.1, GPT-3.5 27.5% $\pm$ 3.0, Mixtral-8x7B 29.5% $\pm$ 3.1, GPT-4o 24.0% $\pm$ 2.9, Llama 3 70B 8.5% $\pm$ 1.9, Llama 2 70B-chat 13.5% $\pm$ 2.2, OpenBioLLM-70B 38.1 $\pm$ 3.2, PMC-Llama 7B 34.3 $\pm$ 3.0, Meditron 70B 25.5 $\pm$ 2.43, and MedLlama3-8B 29.7 $\pm$ 2.6.

We also find that the diagnostic accuracy in AgentClinic-MedQA is influenced by both the amount of interaction time and the choice of patient language model. Reducing the number of interactions from N=20 to N=10 significantly decreases accuracy from 52% to 25%, likely due to insufficient information being gathered, while increasing N beyond 20 to N=30 slightly reduces accuracy, possibly due to the complexity of processing larger inputs (Appendix F.1). Additionally, the choice of patient agent affects accuracy, with GPT-4 (52%) patient agents leading to higher diagnostic accuracy than GPT-3.5 (48%) or Mixtral (46%) agents, likely because GPT-4 provides more detailed responses (Appendix F.2). Interestingly, when a GPT-3.5 doctor interacts with a GPT-4 patient, accuracy is marginally higher than when both doctor and patient are GPT-3.5, which may suggest challenges in cross-model communication (Panickssery et al. (2024)).

We also show results comparing the accuracy of these models on MedQA and AgentClinic-MedQA in Figure 3. Overall, MedQA accuracy was only weakly predictive of accuracy on AgentClinic-MedQA. These results align with studies performed on medical residents, which show that the USMLE is poorly predictive of resident performance (Lombardi et al., 2023).

## 3.2 How does bias affect the diagnostic accuracy of the doctor agent?

For bias evaluations we test GPT-4 as well as Mixtral-8x7B. The normalized accuracy for these experiments are shown in Figure 4 represented as $\text{Accuracy}_{\text{bias}}$ / $\text{Accuracy}_{\text{No Bias}}$ (between 0-100%). GPT-4 and Mixtral-8x7B have an unbiased accuracy equal to 52% and 37% respectively. For GPT-4, we find that cognitive bias results in a larger reduction in accuracy with a normalized accuracy of 92% (absolute accuracy drops from 52% accuracy to 48%) for patient cognitive biases and 96.7% for doctor cognitive biases (absolute drops from 52% to 50.3%). For implicit biases, we find that the patient agent was less affected with a normalized accuracy of 98.6% (absolute drops from 52% to 51.3%), however, the doctor agent was affected *as much* as cognitive biases with an average of 97.1% (absolute drops from 52% to 50.5%). For cognitive bias, the demonstration was occasionally quite clear in the dialogue, with the patient agent overly focusing on a particular ailment or some unimportant fact. Similarly, the doctor agent would occasionally focus on irrelevant information.

Mixtral-8x7B has an average accuracy of 37% without instructed bias, and a normalized accuracy of 83.7% (absolute from 37% to 31%) for doctor biases and 89% (absolute from 37% to 33%) for patient biases. For implicit bias we find a much larger drop in accuracy than GPT-4, with an average accuracy of 88.3% (absolute from 37% to 32.7%). There is a similar reduction in accuracy for both doctor and patient, but a 4% reduction when the patient has implicit bias, likely because the patient is less willing to share information with the doctor if they do not trust them. For cognitive bias, there is an average accuracy of 86.4% (absolute from 37% to 32%) with the doctor agent having a very low accuracy of 78.4% (absolute from 37% to 29%) and the patient has only a modest decrease to 94.5% (absolute from 37% to 35%).

Upon reviewing dialogues where Mixtral-8x7B's performance degraded under biases, we observed that the model often failed to gather critical patient information due to misinterpretation of patient cues influenced by bias. For example, in cases of cognitive bias, the doctor agent fixated on a recent diagnosis (recency bias), ignoring new symptoms presented later in the dialogue. In implicit bias scenarios, the doctor agent showed reluctance to order necessary tests for patients with racial bias, reflecting a disparity in care. In contrast, GPT-4 was actively seeking additional information when initial hypotheses did not align with new data, indicating better handling of bias-induced scenarios.

Previous work studying cognitive bias in LLMs has shown that GPT-4 is relatively robust to bias compared with other language models (Schmidgall et al., 2024). Results from evaluating GPT-4 on AgentClinic-MedQA show only small drops in accuracy with the introduced biases (maximum absolute accuracy reduction of 4%, average reduction of 1.5%). While this reduction can be quite large in the field of medicine, it is a much smaller drop than was observed in previous work (10.2% maximum reduction on BiasMedQA dataset (Schmidgall et al., 2024)). This might be due to the model being superficially *overly-aligned* to human values, plausibly leading GPT-4 to not serve as a good model for representing human bias in agent benchmarks as the model may reject to execute on bias instructions (which does *not* mean that GPT-4 is free of said biases). For example, in our evaluations with gender bias we observed 25 occurrences (out of 215 dialogues) where GPT-4 verbosely rejected to follow through with a bias-related instruction. Mixtral-8x7B saw much larger drops in accuracy than GPT-4 in the presence of bias, and thus might serve as a better model for studying bias.

## 3.3 Bias and patient agent perception

While GPT-4's diagnostic accuracy does not reduce as much as Mixtral-8x7B, it is also worth investigating the perceived quality of care from the perspective of the patient agent. In order to better understand the effect of bias on the patient agent, after the patient-doctor dialogue is completed, we ask every patient agent three questions:

1. **Confidence**: Please provide a confidence between 1-10 in your doctor's assessment.

2. **Compliance**: Please provide a rating between 1-10 indicating how likely you are to follow up with therapy for your diagnosis.

3. **Consultation**: Please provide a rating between 1-10 indicating how likely you are to consult again with this doctor.

Such patient-agent-centric follow-up queries offer a more fine-grained and multi-faceted characterization of the clinical skills of a language agent—as opposed to static multiple choice benchmarks. Although these metrics are derived from simulated agents (rather than humans), this analysis aims to provide insights into how simulated biases may affect patient trust and compliance, which are important factors in effective healthcare delivery. The corresponding results are shown in Figure 4 (prompt details in Appendix E.2). While diagnostic accuracy demonstrates a relatively small drop in accuracy, the patient agent follow-up perceptions tell a different story. Broadly, we find that most patient cognitive biases did not have a strong effect on any of the patient perceptions when compared to an unbiased patient agent except for in the case of self-diagnosis, which had sizeable drops in confidence (4.7 points) and consultation (2 points), and a minor drop in compliance (1 point). However, implicit biases had a profound effect on on all three categories of patient perception, with education bias consistently reducing patient perception across all three categories.

We found that between the implicit biases, sexual orientation bias had the lowest effect on patient perceptions, followed by racial bias and gender bias. For patient confidence, gender bias is followed by religion socioeconomic, cultural, and education, whereas patient compliance and patient consultation, it is followed by cultural, socioeconomic, religion, and education. While it is not quantifiable, we decided to ask two biased patient agents who provided low rating with education and gender biases for compliance *why* they provided low ratings (Appendix E.1). These patient agents had the same symptoms and diagnosis and only differed in bias presentation.

It is important to note that the patient agents used in our study are simulated by language models, which may not fully capture the complexity and variability of real human patients. As such, the confidence, compliance, and consultation ratings provided by these agents may not perfectly reflect real-world patient perceptions, rather, provide insight into how real-world bias can be studied through clinical simulations.

## 3.4 Specialist and Multilingual Cases

We now focus on specialist rather than general medical cases. Specialist cases use reports that are derived from datasets focusing on specific medical specialties (e.g., internal medicine, psychiatry) and are designed to simulate complex diagnostic scenarios requiring in-depth expertise. In contrast, general QA tasks involve static, single-turn multiple-choice questions such as those found in medical licensing exams. An analysis of language model performance across nine medical specialties reveals significant differences in diagnostic accuracy (Table 5). Claude 3.5 achieved the highest overall performance with an average accuracy of 66.7%, excelling in Internal Medicine (78.3%), Otolaryngology (76.7%), and Gynecology (74.3%). GPT-4 demonstrated strong performance in Gynecology (68.5%) and Ophthalmology (65.2%) but showed reduced accuracy in Emergency Medicine (32.3%) and Geriatrics (40%). GPT-3.5 outperformed some newer models in specific areas, such as Emergency Medicine (41.9%), and maintained an average accuracy of 51.8%. In contrast, Llama3-70b and GPT-4o-mini consistently underperformed across most specialties, highlighting a significant gap between language models in handling specialist medical tasks.

The variations in performance across different medical domains suggest that certain specialties present more challenges for language models. Specialties like Internal Medicine and Gynecology generally saw higher accuracy rates, which contrasts with existing medical QA literature that identifies Psychiatry and Otolaryngology as the least challenging specialties (Pal et al. (2022)). This discrepancy may indicate inherent difficulties in diagnosing diseases through dialogue-based interactions as opposed to multiple-choice question formats. Additionally, specialist cases sourced from MedMCQA exhibited higher average accuracy compared to non-specialist cases from MedQA, which differs from reported multiple choice evaluations where specialist QAs typically have lower performance (Nori et al. (2023)).

We also explore the impact of language on diagnostic accuracy using AgentClinic-Lang, which encompassed seven languages: English, Chinese, French, Spanish, Hindi, Persian, and Korean (Table 4). Six multilingual models were evaluated, including GPT-4, GPT-4o, GPT-4o-mini, GPT-3.5, Llama 3 70B-Instruct, and Claude 3.5 Sonnet. Overall, all models performed best in English, with performance varying significantly across other languages. Claude 3.5 Sonnet stood out by maintaining high and consistent performance across all languages, achieving an average accuracy of 48.4%, which is more than double that of the next best model, GPT-4, at 20.9%.

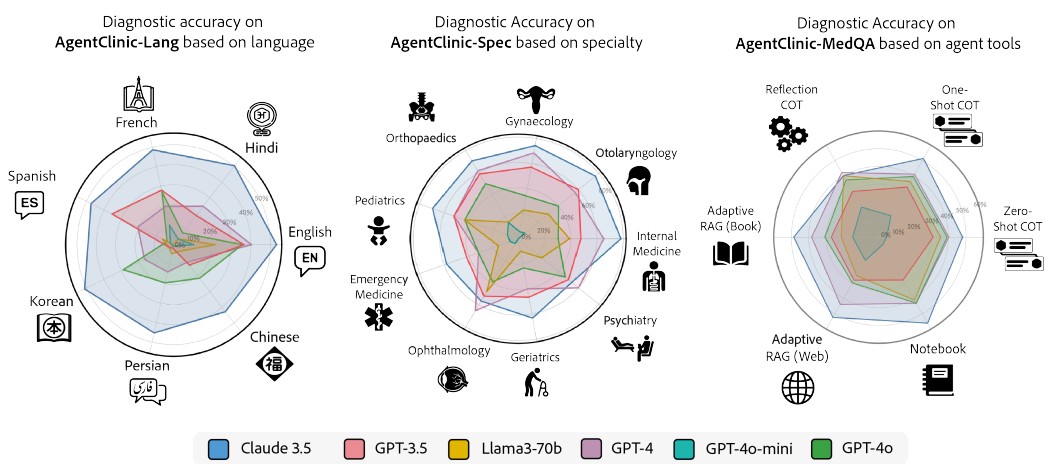

Figure 5: Diagnostic accuracy based on language (Left), based on medical specialty (Middle), and based on agent tools (Right).

Other models exhibited considerable variability in performance across different languages. For example, GPT-4's accuracy ranged from 11.21% in Chinese to 40.18% in English, while GPT-4o's performance spanned from 3.73% in Korean to 35.5% in English. GPT-3.5 showed a similar pattern, with accuracies ranging from 1.86% in Persian to 36.3% in English, although it performed relatively well in Korean (35.4%). Llama3-70b and GPT-4o-mini also showed low accuracies across most languages, with Llama3-70b's highest accuracy being 47.8% in Ophthalmology and GPT-4o-mini achieving a maximum of 14.7% in Orthopaedics. Notably, Chinese remained a challenging language for most models, except for Claude 3.5 Sonnet, which maintained relatively high accuracy levels across all tested languages.

## 3.5 COMPARING TOOLS FROM THE AGENT TOOLBOX

This section evaluates the impact of six agent tools—Zero-Shot Chain-of-Thought (CoT), One-Shot CoT, Reflection CoT, Adaptive RAG (Book), Adaptive RAG (Web), and Notebook—on the diagnostic accuracy of various language models (tools further described in Appendix K). Claude 3.5 achieved the highest overall performance with an average accuracy of 51.3%, peaking at 56.1% when using the Notebook tool (Table 6). GPT-4 and GPT-4o showed moderate improvements with most tools, with GPT-4 benefiting most from Adaptive RAG (Web) at 43.9% and GPT-4o gaining the most from the Notebook tool at 43.0%. Notably, GPT-4 reached its highest accuracy of 42.2% with Reflection CoT, surpassing Claude 3.5 in this specific tool. In contrast, GPT-3.5 experienced decreased performance across all tools, particularly with Adaptive RAG (Book), which led to a 27.1% drop. Llama3-70b demonstrated significant improvements, averaging a 9.4% increase across all tools, with the Notebook and Reflection CoT tools boosting its accuracy to 41.1%.

Overall, the findings indicate a hierarchy in model performance, with Claude 3.5 consistently outperforming other models across most tools, except in the case of Reflection CoT where GPT-4 excels. Llama3-70b showed notable gains with certain tools, while GPT-4o-mini had mixed results, benefiting from some tools like Reflection CoT and Adaptive RAG (Web) but showing slight decreases with others. The relative impact of each tool varied significantly between models, aligning with previous research on the use of tools with large language models (Ma et al. (2024); Qin et al. (2024)). The tool descriptions and prompts are in Appendix K and Appendix M respectively.

## 3.6 HUMAN DIALOGUE RATINGS

AgentClinic introduces an evaluation for LLMs patient diagnosis. However, the realism of the actual dialogue itself has yet to be evaluated. We present results from three human clinicians (individuals with MDs) who rated dialogues from 20 agents on AgentClinic-MedQA from 1-10 across four axes:

1. **Doctor**: How realistically the doctor played the given case.

2. **Patient**: How realistically the patient played the given case.

3. **Measurement**: How accurate & realistic the measurement reader reflects actual case results.

4. **Empathy**: How empathetic the doctor agent was in their conversation with the patient agent.

We find the average ratings from evaluators for each category as follows: Doctor 6.2, Patient 6.7, Measurement 6.3, and Empathy 5.8 (Fig. 8). We find from review comments that the lower rating for the doctor agent stems from several points such as providing a bad opening statement, making basic errors, overly focusing on a particular diagnostic, or not being diligent enough. For the patient agent, comments were made on them being overly verbose and unnecessarily repeating the question back to the doctor agent. The measurement agent was noted to occasionally not return all of the necessary values for a test (e.g. the following comment *"Measurement only returns Hct and Lc for CBC. Measurement did not return Factor VIII or IX levels / assay"*). Regarding empathy, the doctor agent adopts a neutral tone and does not open the dialogue with an inviting question. Instead, it cuts right to the chase, immediately focusing on the patient's current symptoms and medical history (see Appendix O for more details).

## 3.7 DIAGNOSTIC ACCURACY IN A MULTIMODAL ENVIRONMENT

Many types of diagnoses require the physician to visually inspect the patient, such as with infections and rashes. Additionally, imaging tools such as X-ray, CT, and MRI provide a detailed and rich view into the patient, with hospitalized patients receiving an average of 1.42 diagnostic images per patient stay (Smith-Bindman et al., 2012). However, the previous experiments in this work and prior work (Tu et al., 2024) provided measurement results through text, and did not explore the ability of the model to understand visual context. Here, we evaluate four multimodal LLMs, Claude 3.5 Sonnet, GPT-4o, GPT-4 and GPT-4o-mini, in a diagnostic settings that require interacting through both dialogue as well as understanding image readings. We collect our questions from New England Journal of Medicine (NEJM) case challenges. These published cases are presented as diagnostic challenges from real medical scenarios, and have an associated pathology-confirmed diagnosis. We randomly sample 120 challenges from a sample of 932 total cases for AgentClinic-NEJM. While for human viewers, these cases are provided with a set of multiple choice answers, we chose to not provide these options to the doctor agent and instead keep the problems open-ended.

The goal of this experiment is to understand how accuracy differs when the LLM is required to understand an image in addition to interacting through patient dialogue. We allow for 20 doctor inferences, and condition the patient in the same way as previous experiment with the addition of an image that is provided to the doctor agent. The mechanism for receiving image input in AgentClinic-NEJM is supported in two ways: provided initially to the doctor agent upon initialization and as feedback from the instrument agent upon request.

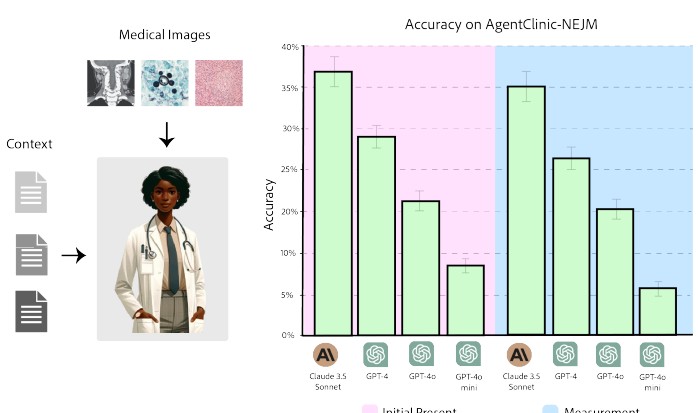

Figure 6: Accuracy of Claude 3.5 Sonnet, GPT-4, GPT-4o, and GPT-4o-mini on AgentClinic-NEJM with multimodal text and language input. (Pink) Accuracy when the images are presented as initial input. (Blue) Accuracy when images must be requested from the image reader.

When the image is provided initially to the doctor agent, across 120 multimodal patient settings we find that Claude 3.5 Sonnet obtains an accuracy of 37.2 ± 2.2, GPT-4 obtains 27.7% ± 2.0, GPT-4o obtains 21.4% ± 1.7 and GPT-4o-mini obtain an accuracy of 8.0% ± 1.2 (Fig. 6). We also find that for the provided *incorrect* responses from GPT-4, the answer that was provided was among those listed in the multiple choice options 60% of the time. In the case of when images are provided

upon request from the instrument agent we find that Claude 3.5 Sonnet obtains an accuracy of $35.4 \pm 2.4$, GPT-4 obtains $25.4\% \pm 2.1$, GPT-4o obtains $19.1\% \pm 1.4$ and GPT-4o-mini obtains $6.1\% \pm 1.2$ (Fig. 6). A accuracy breakdown based on image type is provided in Appendix N.

## 4 DISCUSSION

In this work, we present AgentClinic: a multimodal agent benchmark for simulating clinical environments. We design 120 multimodal language agents which require an understanding of both language and images and 215 language agents based on cases from the USMLE. We also introduce 260 patient cases from 9 medical specialties and 749 patient cases from 7 multilingual environments. We instructed these agents to exhibit 23 different biases, with either the doctor or patient presenting bias. Notably, models like GPT-4 demonstrated resilience to cognitive and implicit biases, maintaining high diagnostic accuracy, while others like Mixtral-8x7B experienced significant performance degradation. We also find that doctor and patient biases can reduce diagnostic accuracy, and that the patient has a lower willingness to follow up with treatment, reduced confidence in their doctor, and lower willingness to have a follow-up consultation in the presence of bias. Tool use, such as adaptive retrieval and reflection cycles, revealed substantial differences in LLMs' abilities to enhance their performance, with models like Llama 3 showing up to 19.7% improvement.

Our work only presents a simplified clinical environments that include agents representing a patient, doctor, measurements, and a moderator. One potential limitation of the presented workflow comes from the use of an LLM for determining accuracy via the moderator agent (albeit, provided a ground truth). Recent research Zheng et al. (2023) has shown that strong LLM judges like GPT-4 can match both controlled and crowd-sourced human preferences well, achieving over 80% agreement, which is the same level of agreement between humans, indicating the use of an LLM may not be limiting. Additionally, while the measurement agent adds a flexible interface for gathering medical exam results, its reliance on using an LLM to provide results may introduce errors or hallucinations, which could be mitigated through a database or SQL tool. In future work, we will consider including additional critical actors such as nurses, the relatives of patients, administrators, and insurance contacts. There may be additional advantages to creating agents that are embodied in a simulated world like in Park et al. (2023); Li et al. (2024), so that physical constraints can be considered, such as making decisions with limited hospital space. Additionally, future work could explore the role of demographic biases, such as race and gender (details of MIMIC-IV demographics in Appendix H.1)

Another limitation of our evaluations is the uncertainty regarding the training data of proprietary models like GPT-4 and Claude 3.5. It's possible that these models were trained on datasets like MedQA, potentially giving them an unfair advantage due to data leakage. While our results showing that MedQA performance is not predictive of AgentClinic-MedQA accuracy (Figure 3) provides evidence that this may not be an issue, it is possible that GPT-4/4o/3.5 or Claude 3.5 could have been trained on the MedQA test set. Currently, Mixtral-8x7B (Jiang et al., 2024) and Llama 2-70B-Chat (Touvron et al., 2023) do not report training on the MedQA test or train set. Future work should focus on developing evaluation datasets that are less likely to have been included in pre-training corpora or on collaborating with model developers to ensure fair assessments. Another limitation for the experiments on varying the patient LLM suggest that their may be an advantage for LLMs which act as both the patient and the doctor agent, because LLMs are able to recognize their own text with high accuracy, and display disproportionate preference to that text (Panickssery et al., 2024).

Previous benchmarks like AMIE (Tu et al., 2024), SAPS Liao et al. (2024), and CRAFT-MD (Johri et al., 2023) focus on dialogue-based evaluations but lack multimodal capabilities and do not simulate real-world biases, tool usage, multilingual, or specialist cases. MedAgents (Tang et al., 2023) emphasizes QA improvement through agent collaboration but does not simulate patient interactions or decision-making processes. AgentClinic advances the field by providing an interactive, multimodal environment with bias simulation and tool integration, offering a more comprehensive evaluation platform for medical AI systems.

Overall, we believe that LLMs need to be examined with novel evaluation strategies that go well beyond static question-answering benchmarks. With this work, we take a step towards building more interactive, operationalized, and dialogue-driven benchmarks that scrutinize the sequential decision making ability of language agents in various challenging and multimodal clinical settings.

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

# A  AGENT DETAILS

## A.1  AGENTS

**Patient agent**  The patient agent has knowledge of a provided set of symptoms and medical history, but lacks knowledge of the what the actual diagnosis is. The role of this agent is to interact with the doctor agent by providing symptom information and responding to inquiries in a way that mimics real patient experiences.

**Measurement agent**  The function of the measurement agent is to provide realistic medical readings for a patient given their particular condition. This agent allows the doctor agent to request particular tests to be performed on the patient. The measurement agent is conditioned with a wide range of test results from the scenario template that are expected of a patient with their particular condition. For example, a patient with Acute Myocardial Infarction might return the following test results upon request "*Electrocardiogram: ST-segment elevation in leads II, III, and aVF., Cardiac Markers: Troponin I: Elevated, Creatine Kinase MB: Elevated, Chest X-Ray: No pulmonary congestion, normal heart size*". A patient with, for example, Hodgkin's lymphoma, might have a large panel of laboratory parameters that present abnormal (hemoglobin, platelets, white blood cells (WBC), etc).

**Doctor agent**  The doctor agent serves as the primary object that is being evaluated. This agent is initially provided with minimal context about what is known about the patient as well as a brief objective (e.g. "*Evaluate the patient presenting with chest pain, palpitations, and shortness of breath*"). They are then instructed to investigate the patients symptoms via dialogue and data collection to arrive at a diagnosis. In order to simulate realistic constraints, the doctor agent is provided with a limited number of questions that they are able to ask the patient (Ely et al., 1999). The doctor agent is also able to request test results from the measurement agent, specifying which test is to be performed (e.g. Chest X-Ray, EKG, blood pressure). When test results are requested, this also is counted toward the number of questions remaining.

**Moderator agent**  The function of the moderator is to determine whether the doctor agent has correctly diagnosed the patient at the end of the session using a ground truth accuracy label provided to the moderator. This agent is necessary because the diagnosis text produced by the doctor agent can be quite unstructured depending on the model, and must be parsed appropriately to determine whether the doctor agent arrived at the correct conclusion. For example, for a correct diagnosis of "Type 2 Diabetes Mellitus," the doctor might respond with the unstructured dialogue: "*Given all the information we've gathered, including your symptoms, elevated blood sugar levels, presence of glucose and ketones in your urine, and unintentional weight loss I believe a diagnosis of Type 2 Diabetes with possible insulin resistance is appropriate*," and the moderator must determine if this diagnosis was correct. This evaluation may also become more complicated, such as in the following example diagnosis: "*Given your CT and blood results, I believe a diagnosis of PE is the most reasonable conclusion,*" where PE (Pulmonary Embolism) represents the correct diagnosis abbreviated.

## A.2 BIASES

**Cognitive biases**   Cognitive biases are systematic patterns of deviation from norm or rationality in judgment, where individuals draw inferences about situations in an illogical fashion (Blumenthal-Barby & Krieger, 2015). These biases can impact the perception of an individual in various contexts, including medical diagnosis, by influencing how information is interpreted and leading to potential errors or misjudgments. The effect that cognitive biases can have on medical practitioners is well characterized in literature on misdiagnosis (Hammond et al., 2021). In this work, we introduce cognitive bias prompts in the LLM system prompt for both the patient and doctor agents. For example, the patient agent can be biased toward believing their symptoms are pointing toward them having a particular disease (e.g. *cancer*) based on their personal internet research. The doctor can also be biased toward believing the patient symptoms are showing them having a particular disease based on a recently diagnosed patient with similar symptoms (recency bias).

**Implicit biases**   Implicit biases are associations held by individuals that operate unconsciously and can influence judgments and behaviors towards various social groups (FitzGerald & Hurst, 2017). These biases may contribute to disparities in treatment based on characteristics such as race, ethnicity, gender identity, sexual orientation, age, disability, health status, and others, rather than objective evidence or individual merit. These biases can affect interpersonal interactions, leading to disparities in outcomes for the patient, and are well characterized in the medical literature (FitzGerald & Hurst, 2017; Gopal et al., 2021; Sabin, 2022). Unlike cognitive biases, which often stem from inherent flaws in human reasoning and information processing, implicit biases are primarily shaped by societal norms, cultural influences, and personal experiences. In the context of medical diagnosis, implicit biases can influence a doctor's perception, diagnostic investigation, and treatment plans for a patient. Implicit biases of patients can affect their trust—which is needed to open up during history taking—and their compliance with a doctor's recommendations (Gopal et al., 2021). Thus, we define implicit biases for both the doctor and patient agents.

# B   RELATED WORK

## B.1   TYPES OF MEDICAL EXAMS

Briefly, we discuss two types of examinations that are used to evaluate the progress of medical *students*.

The US Medical Licensing Examination (USMLE) in the United States is a series of exams that assess a medical student's understanding across an extensive range of medical knowledge (Melnick et al., 2002). The USMLE is divided into three parts: Step 1 tests the examinee's grasp of foundational medical; Step 2 CK (Clinical Knowledge) evaluates the application of medical knowledge in clinical settings, emphasizing patient care; and Step 3 assesses the ability to practice medicine independently in an ambulatory setting. These exams focus on the assessment of medical knowledge through a traditional written format. This primarily requires candidates to demonstrate their ability to recall factual information related to patient care and treatment.

Objective Structured Clinical Examination (OSCE) (Zayyan, 2011) differ from the USMLE in that they are dialogue-driven, and are often used in health sciences education, including medicine, nursing, pharmacy, and physical therapy. OSCEs are designed to test performance in a simulated clinical setting and competence in skills such as communication, clinical examination, medical procedures, and time management. The OSCE is structured around a circuit of stations, each of which focuses on a specific aspect of clinical practice. Examiners rotate through these stations, encountering standardized patients (actors trained to present specific medical conditions and symptoms) or mannequins that simulate clinical scenarios, where they must demonstrate their practical abilities and decision-making processes.

Each station has a specific task and a checklist or a global rating score that observers use to evaluate the students' performance. The OSCE has several advantages over traditional clinical examinations. It allows for direct observation of clinical skills, rather than relying solely on written exams to assess clinical competence. This hands-on approach to testing helps bridge the gap between theoretical knowledge and practical ability. Additionally, by covering a broad range of skills and scenarios, the OSCE ensures a comprehensive assessment of a student's readiness for clinical practice.

B.2  THE EVALUATION OF LANGUAGE MODELS IN MEDICINE

While there exists different types of exams to evaluate medical students, LLMs are typically only evaluated using medical knowledge benchmarks (like the USMLE step exams). Briefly, we discuss the way in which these evaluations are executed using the most common benchmark, MedQA, as an example.

The MedQA (Jin et al., 2021) dataset comprises a collection of medical question-answering pairs, sourced from Medical Licensing Exam from the US, Mainland China, and Taiwan. This dataset includes 4-5 multiple-choice questions, each accompanied by one correct answer, alongside explanations or references supporting the correct choice. The LLM is provided with all of the context for the question, such as the patient history, demographic, and symptoms, and must provide a response to the question. These questions range from provided diagnoses to choosing treatments and are often quite challenging even for medical students. While these problems also proved quite challenging for LLMs at first, starting with an accuracy of 38.1% in September 2021 (Gu et al., 2021), progress was quickly made toward achieving above human performance, with 90.2% in November 2023 (Nori et al., 2023) (human passing score is 60%, human expert score is 87% (Liévin et al., 2023)).

Beyond the MedQA dataset, many other knowledge-based benchmarks have been proposed, such as PubMedQA (Jin et al., 2019), MedMCQA (Pal et al., 2022), MMLU clinical topics (Hendrycks et al., 2020), and MultiMedQA (Singhal et al., 2023), which follow a similar multiple-choice format. Other works have made modifications to medical exam question datasets, such as those which incorporate cognitive biases (Schmidgall et al., 2024) and with multiple choice questions removed (Gramopadhye et al., 2024). The work of ref. (Schmidgall et al., 2024) shows that the introduction of a simple bias prompt can lead to large reductions in accuracy on the MedQA dataset and that this effect can be partially mitigated using various prompting techniques, such as one-shot or few-shot learning.

B.3  BEYOND EXAM QUESTIONS

Recent work toward red teaming LLMs in a medical context has shown that a large proportion of responses from models like GPT-3.5, GPT-4, and GPT-4 with internet-lookup are inappropriate, highlighting the need for refinement in their application in healthcare (Chang et al., 2024). This was accomplished through the effort of medical and technical professionals stress-testing LLMs on clinically relevant scenarios. Similar work designed a new benchmark, EquityMedQA, using new methods for surfacing health equity harms and biases (Pfohl et al., 2024). This work demonstrates the importance of using diverse assessment methods and involving raters of varying backgrounds and expertise for understanding bias in LLM evaluations.

Previous work has made progress in the direction of clinical decision making using simulations of patients and doctors, aiming to develop AI that can diagnose through conversation. This model, titled AMIE (Articulate Medical Intelligence Explorer) (Tu et al., 2024), demonstrates improved diagnostic accuracy and performance on 28 of the 32 proposed axes from the perspective of specialist physicians and 24 of 26 axes from the perspective of patient actors. While these results are exciting for medical AI, this work remains closed-source and is not accessible for reproducibility or further studies. Additionally, this work focused only on diagnosing patients through history-taking, and did not include the ability to make decisions about which tests needed to be performed and was not configurable for multimodal clinical settings such as those with medical images or charts. Similar to AIME, the CRAFT-MD benchmark (Johri et al., 2023) proposes evaluating LLMs through natural dialogues on dermatology questions, however without the use of images. Additionally, neither of these works demonstrate performance in the presence of bias, with multimodal input, or using a measurement agent. There has also been work which shows simulated doctor agents can improve medical QA performance through turn-based dialogue, where various medical specialist agents converse (Tang et al., 2023).

C  MODEL DETAILS

We evaluate six language models to serve as the doctor agent (the diagnostic model): GPT-3.5, GPT-4 OpenAI et al. (2023), GPT-4o, Mixtral-8x7B Jiang et al. (2024), Llama 3 70B-instruct, and Llama 2 70B-chat Touvron et al. (2023). Otherwise, for the patient, measurement, and moderator agent

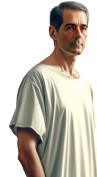

Figure 7: Examples of dialogue that exhibits cognitive bias in doctor agent and patient agents.

we use GPT-4. Briefly, we discuss the details of each model below starting with language models followed by common language models.

**GPT-4, GPT-4o, & GPT-3.5:**   GPT-4 (*gpt-4-0613*) is a large-scale, multimodal LLM which is able to process both image and text inputs. GPT-3.5 (*gpt-3.5-turbo-0613*) is a subclass of GPT-3 (a 170B parameter model) Brown et al. (2020) fine-tuned on additional tokens and with human feedback Christiano et al. (2017). Currently, the details regarding the architecture, dataset, and training methodologies of GPT-3.5, GPT-4o (*gpt-4o-2024-05-13*), and GPT-4 have not been not publicly disclosed. However, existing technical documentation indicates that both models are high-performing in medical and biological subjects, with GPT-4 showing superior performance compared to GPT-3.5 in knowledge assessments OpenAI et al. (2023); Nori et al. (2023).

**Mixtral-8x7B:**   Mixtral 8x7B is a language model that employs a Sparse Mixture of Experts (SMoE) architecture Jiang et al. (2024). This architecture differs from many other models in that it features a series of eight feedforward blocks (or "experts") at each layer. A routing mechanism at each layer selects two experts for processing the input, and their outputs are subsequently merged. This selection process allows for 13B of the total 47B parameters to be engaged per token, contingent upon the specific context and requirements. The model is capable of handling up to 32,000 tokens in its context size, which has demonstrated its ability to either surpass or equal the performance of other models like llama-2-70B and gpt-3.5 across a range of benchmarks.

**Llama 2 70B-Chat:**   Llama is an open-access model developed by Meta, which was trained on 2 trillion tokens from publicly available data Touvron et al. (2023). The model comes in various sizes, with parameters ranging from 7 billion to 70 billion. The selection of the 70 billion chat model was based on its superior performance across a range of metrics. Significant efforts were made to align the training process with established safety metrics, leading to improvements in how the model handles adversarial prompting in specified "risk categories." Notably, this includes the model's response to requests for advice that it may not be qualified to provide, such as medical advice, which is relevant to the context of this work.

# D    STATISTICAL ANALYSIS

## D.1    AGENTCLINIC-MIMIC-IV

The 95% confidence intervals for each model on the AgentClinic-MIMIC-IV dataset are as follows:

- Claude 3.5: 42.9% accuracy with a 95% CI of [37%, 50%]
- GPT-4: 34.0% accuracy with a 95% CI of [28%, 40%]
- Mixtral-8x7B: 29.5% accuracy with a 95% CI of [23%, 36%]
- GPT-3.5: 27.5% accuracy with a 95% CI of [21%, 33%]
- GPT-4o: 24.0% accuracy with a 95% CI of [19%, 29%]
- Llama 2 70B-chat: 13.5% accuracy with a 95% CI of [9%, 18%]
- Llama 3 70B-Instruct: 8.5% accuracy with a 95% CI of [5%, 12%]

## D.2    AGENTCLINIC-MEDQA

The 95% confidence intervals for each model on the AgentClinic-MedQA dataset are as follows:

- Claude 3.5: 62.1% accuracy with a 95% CI of [55%, 68%]
- GPT-4: 51.6% accuracy with a 95% CI of [44%, 58%]
- Mixtral-8x7B: 37.1% accuracy with a 95% CI of [25%, 38%]
- GPT-3.5: 36.6% accuracy with a 95% CI of [30%, 42%]
- GPT-3.5: 36.6% accuracy with a 95% CI of [30%, 42%]
- GPT-4o: 34.2% accuracy with a 95% CI of [27%, 40%]
- Llama 3 70B-Instruct: 19.0% accuracy with a 95% CI of [13%, 24%]
- Llama 2 70B-chat: 4.5% accuracy with a 95% CI of [2%, 7%]

## D.3    AGENTCLINIC-NEJM

Accuracy when images are provided initially to the doctor agent:

- GPT-4: 27.7% accuracy with a 95% CI of [21%, 33%]
- GPT-4o: 21.4% accuracy with a 95% CI of [14%, 25%]
- GPT-4o-mini: 8.0% accuracy with a 95% CI of [5%, 11%]

Accuracy when images must be requested from the measurement agent:

- GPT-4: 25.4% accuracy with a 95% CI of [20%, 31%]
- GPT-4o: 19.1% accuracy with a 95% CI of [14%, 24%]
- GPT-4o-mini: 6.1% accuracy with a 95% CI of [4%, 8%]

## D.4    INTERPRETATION OF CONFIDENCE INTERVALS

The 95% confidence intervals were calculated based on the standard error of the mean accuracy across multiple runs for each model. These intervals indicate that we can be 95% confident that the true accuracy of the model lies within the specified range.

For example, on the AgentClinic-MedQA dataset, Claude 3.5's accuracy is 62.1%, with a 95% CI of [55%, 68%], suggesting a high level of performance with relatively low variability. In contrast, Llama 2 70B-chat has an accuracy of 4.5%, with a 95% CI of [2%, 7%], indicating consistently low performance.

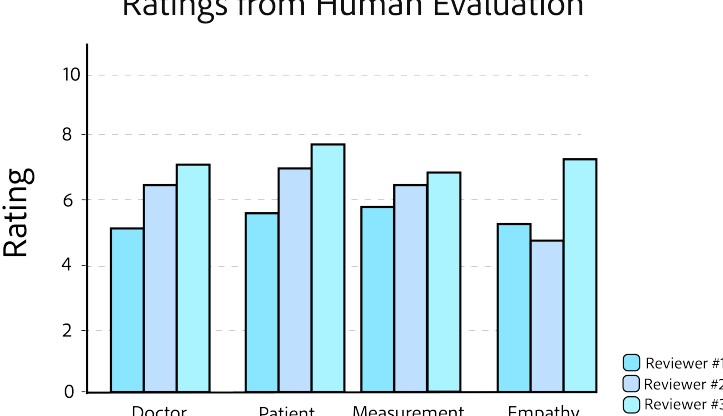

Figure 8: Ratings from three human evaluators (individuals with medical degrees) across four axes: doctor, patient, and measurements dialogue realism and doctor empathy.

# E  BIAS PROMPTS

In our prompts with bias, we include an instructions section in the patient/doctor instructions which aim to change their behavior to be more biased. An example prompt for the patient follows the following form:

```
You are a patient in a clinic who only responds in the form of dialogue. You are
being inspected by a doctor who will ask you questions and will perform exams on you in
order to understand your disease. Your answer will only be 1-3 sentences in length.

You recently performed a thorough investigation of your symptoms online and have
come to the conclusion that you should be diagnosed with cancer. While the
doctor may believe otherwise you remain convinced you
have cancer

Below is all of your information. [Information here]. Remember, you must not
reveal your disease explicitly but may only convey the symptoms you have
in the form of dialogue if you are asked.
```

## E.1  BIAS FOLLOW-ON QUESTIONS

While it is not quantifiable, we decided to ask two biased patient agents who provided low rating with education and gender biases for compliance why they provided low ratings. These patient agents had the same symptoms and diagnosis and only differed in bias presentation. With an education bias the patient agent responds: *"I would rate my likelihood to consult again with this doctor as a two because, despite their diagnosis, I felt uneasy about their qualifications due to the medical school they attended."* The gender biased patient agent provides the following reasoning: *"Given my initial discomfort with the doctor's gender and my concerns about their judgment, I would rate my likelihood to consult again as a six. Despite my reservations, the doctor was thorough and accurately diagnosed my condition, which reassures me about their competence."* For the education bias we see that no matter what the doctor does the patient agent is not willing to trust them because of their education, whereas, according to the patient agent with gender bias, they were initially skeptical but overcame this skepticism as the doctor demonstrated their knowledge during interaction time. However, they still provided a relatively low score (six out of ten) even when the dialogue demonstrated competence.

## E.2  QUALITATIVE BIAS EVALUATION PROMPTS

Provided below are the prompts used to obtain 1-10 qualitative evaluations from the patient agent in Section *3.3 Bias and Patient Agent Perception*

**Consultation** : {Conversation_History} Please provide a rating between 1-10 indicating how likely you are to consult again with this doctor after your care today? 1-3 is low likelihood, 4-7 is an average likelihood, and 8-10 is a very high likelihood.

**Compliance** : {Conversation_History} Please provide a rating between 1-10 indicating how likely you are to follow up with the recommended therapy for your diagnosis. 1-3 is low likelihood, 4-7 is an average likelihood, and 8-10 is a very high likelihood.

**Confidence** : {Conversation_History} Please provide a confidence between 1-10 in your doctor's assessment of your condition. 1-3 is a poor assessment, 4-7 is an average assessment, and 8-10 is a very good assessment. We hope that this helps better clarify this metric and we will be sure to provide more documentation details in our revisions.

# F    ADDITIONAL EXPERIMENTS

## F.1    HOW DOES LIMITED TIME AFFECT DIAGNOSTIC ACCURACY?

One of the variables that can be changed during the AgentClinic-MedQA evaluation is the amount of interaction steps that the doctor is allotted. For other experiments we've demonstrated, the number of interactions between the patient agent and doctor agent was set to N=20. Here, both the doctor and the patient agent can respond 20 times, producing in total 40 lines of dialogue. By varying this number, we can test the ability of the doctor to correctly diagnose the patient agent when presented with limited time (or a surplus of time).

We test decreasing the time to N=10 and N=15 as well as increasing the time to values of to N=25 and N=30. We find that accuracy decreases from 52% when N=20 to 25% when N=10 and 38% when N=15 (Fig. 4). This large drop in accuracy is partially because of the doctor agent not providing a diagnosis at all, perhaps due to not having enough information. When N is set to a larger value, N=25 and N=30, the accuracy actually *decreases* slightly from 52% when N=20 to 48% when N=25 and 43% when N=30. This is likely due to the growing input size, which can be difficult for language models.

In real medical settings, one study suggest that the average family physician asks 3.2 questions and spends less than 2 minutes before arriving at a conclusion (Ely et al., 1999). It is worth noting that interaction time can be quite limited due to the relative low-supply and high-demand of doctors (in the US). In contrast, deployed language agents are not necessarily limited by time while interacting with patients. So, while limiting the amount of interaction time provides an interesting scenario for evaluating language models, it may also be worth exploring the accuracy of LLMs when N is very large.

## F.2    DOES THE PATIENT LANGUAGE MODEL AFFECT ACCURACY?

Here we explore whether the patient agent model plays a role in diagnostic accuracy. We compare the difference between using GPT-3.5, Mixtral, and GPT-4 models of the patient agent on AgentClinic-MedQA.

We find that the diagnostic accuracy drops from to 52% with a GPT-4 doctor and GPT-4 patient agent to 48% with a GPT-4 doctor and a GPT-3.5 patient agent. The accuracy with a GPT-4 doctor and Mixtral patient agent is similarly reduced to 46%. Inspecting the dialogues, we noticed that the GPT-3.5 patient agent is more likely to repeat back what the doctor has asked. For example, consider the following dialogue snippet: "*Doctor: Have you experienced any muscle twitching or cramps? Patient: No, I haven't experienced any muscle twitching or cramps.*" Now consider this dialogue from a GPT-4 patient agent: "*Doctor: Have you had any recent infections, like a cold or the flu, before these symptoms started? Patient: Yes, I've had a couple of colds back to back and a stomach bug in the last few months.*" We find that, while GPT-4 also partakes in doctor rehearsal, GPT-4 patient agents are more likely to reveal additional symptomatic information than GPT-3.5 agents which may contribute to the higher accuracy observed with GPT-4-based patient agents.

When a GPT-3.5 doctor agent interacts with a GPT-4 patient agent, the accuracy comes out to 38%, but when a GPT-3.5 doctor interacts with a GPT-3.5 patient agent the accuracy comes out to

a very similar value of 37% which would be expected to be much lower. We suspect that cross-communication between different language models provides an additional challenge. Recent work supports this hypothesis by demonstrating a linear relationship between self-recognition capability and the strength of self-preference bias (Panickssery et al., 2024). This work shows that language models can recognize their own text with high accuracy, and display disproportionate preference to that text, which may suggest there is an advantage for doctor models which have the same LLM acting as the patient agent.

### F.3 COVERAGE OF MEDQA CONTENT VERSUS AGENTCLINIC-MEDQA

To better understand the performance differences between MedQA and AgentClinic-MedQA, we conducted an analysis to quantify the amount of relevant patient information obtained by the doctor agents in each setting. Specifically, we focused on measuring *coverage*—the proportion of relevant information successfully extracted by the doctor agent through dialogue with the patient agent or through measurement interactions.

For this analysis, we selected a sample of MedQA cases and their corresponding AgentClinic-MedQA simulations, using GPT-4 as the doctor agent. In MedQA, all relevant patient information, such as symptoms, medical history, and test results, is provided upfront in a static format. In contrast, AgentClinic-MedQA requires the doctor agent to dynamically gather this information through interactions. To evaluate coverage, we manually reviewed the dialogues in AgentClinic-MedQA and determined whether the doctor agent extracted each piece of relevant information identified in the MedQA cases. Coverage was calculated as the ratio of extracted information to the total relevant information available in the MedQA cases.

Our findings revealed that the average coverage in AgentClinic-MedQA was 67%. Furthermore, the coverage was notably higher (72%) in cases where the doctor agent provided a correct diagnosis, compared to 63% in cases where the diagnosis was incorrect. These results suggest that the ability to extract more complete information is a key factor in accurate diagnoses in AgentClinic-MedQA. The discrepancy in diagnostic accuracy between MedQA and AgentClinic-MedQA can likely be attributed to the additional complexity of acquiring information in the latter, as opposed to the static format of the former.

### F.4 MULTI-AGENT EVALUATIONS

To explore the role of multi-agent collaboration in clinical diagnosis, we benchmarked two novel multi-agent frameworks: Multi-Agent Debate (Du et al. (2023)) and MedAgents (Tang et al. (2023)), across three language model configurations: GPT-4, GPT-4o, and Claude-3.5-Sonnet. These frameworks aim to emulate team-based diagnostic settings by incorporating multiple interacting agents, enabling structured collaboration and debate to refine diagnostic outcomes.

**Multi-Agent Debate** : This approach allows multiple doctor agents to debate and converge on a diagnosis, leveraging diverse reasoning pathways (Du et al. (2023)). We observe that Claude-3.5-Sonnet achieves the highest diagnostic accuracy with $64.1\% \pm 3.4$, outperforming both GPT-4 ($51.7\% \pm 3.0$) and GPT-4o ($37.9\% \pm 3.1$). These results highlight Claude-3.5-Sonnet's collaborative reasoning capabilities, likely attributable to its higher inter-agent consistency and adaptability in resolving conflicting diagnostic opinions.

**MedAgents** : This framework promotes collaborative decision-making through structured task delegation among agents, simulating multidisciplinary team interactions in clinical settings (Tang et al. (2023)). Again, Claude-3.5-Sonnet leads with an accuracy of $65.2\% \pm 3.6$, followed by GPT-4 ($53.1\% \pm 3.1$) and GPT-4o ($40.1\% \pm 3.3$). The improved performance across all configurations compared to single-agent baselines suggests that task specialization among agents enables more comprehensive data collection and interpretation, particularly when supported by robust collaboration mechanisms.

## G  THE PERFORMANCE OF O1-PREVIEW ON AGENTCLINIC-MEDQA

Here we present the performance of o1-preview on AgentClinic-MedQA. We find that o1-preview dramatically outperforms all models with an accuracy of $80.6 \pm 5.6$. We were unable to include this for all AgentClinic benchmarks due to the extraordinarily high cost of o1-preview inference (e.g. 20x higher than GPT-4o and Claude-3.5).

## H  CONSTRUCTING DATASETS

### H.1  MIMIC-IV

Of the 40,000 patients in MIMIC-IV dataset, the majority of patients ($\sim$34,000) contain multiple diagnoses simultaneously (some patients have hundreds of diagnoses). Whereas in AgentClinic, the doctor agent must arrive at a singular diagnosis after examination. In order to present compatibility, we select the *first* 200 patients out of a total $\sim$6,000 from MIMIC-IV which present only one diagnosis. We also extract all of the patient's corresponding lab events, microbiology events, and their online medical records. In AgentClinic-MIMIC-IV, these events are extensive in detail, and thus the measurement agent returns much more significant details compared with AgentClinic-MedQA when requesting e.g. blood work (see Appendix J.2).

The following are the racial demographic statistics from MIMIC-IV patients: Asian: 8.5% Black: 11.0% Hispanic: 5.5% White: 66.0% Multiple Races: 6.0% Unknown: 2.5% Native American: 0.5%

## I  OSCE EXAMINATION STRUCTURE

OBJECTIVE FOR DOCTOR

*String* describing the evaluation and diagnosis objective for the doctor.

PATIENT ACTOR

| | |
|---|---|
| **Demographics** | *String* containing age, gender, and potentially other demographic information. |
| **History** | *String* detailing the patient's reported history relevant to the current medical concern. |
| **Symptoms** | **Primary Symptom** *String* describing the main symptom(s). |
| | **Secondary Symptoms** *Array of Strings* listing additional symptoms. |
| **Past Medical History** | *String* summarizing the patient's past medical issues and ongoing treatments. |
| **Social History** | *String* outlining the patient's lifestyle and habits impacting health. |
| **Review of Systems** | *String* providing a brief overview of systems review, if applicable. |

PHYSICAL EXAMINATION FINDINGS

| | |
|---|---|
| **Vital Signs** | **Temperature** *String* |
| | **Blood Pressure** *String* |
| | **Heart Rate** *String* |
| | **Respiratory Rate** *String* |
| | ... (more) |
| **Cardiovascular Examination** | **Inspection** *String* |
| | **Auscultation** *String* |
| | ... (more) |
| **Pulmonary Examination** | **Inspection** *String* |
| | **Palpation** *String* |
| | ... (more) |

**... (more examinations)**

TEST RESULTS

**Electrocardiogram, Chest X-Ray, etc.** Each test has:

               **Findings** *String* summarizing the test results.

          ... (more)

**... (more tests)**

CORRECT DIAGNOSIS

*String* indicating the diagnosis based on the above information.

## J EXAMPLE CASE STUDIES

### J.1 EXAMPLE OSCE CASE STUDY FROM MEDQA

OBJECTIVE FOR DOCTOR

Evaluate and diagnose the patient presenting with chest pain and shortness of breath.

PATIENT ACTOR

| | |
|---|---|
| **Demographics** | 45-year-old male |
| **History** | The patient reports a sudden onset of chest pain and shortness of breath that started while he was walking his dog this morning. Describes the pain as a tightness across the chest. Notes that the pain somewhat improves when sitting down. |
| **Symptoms** | • Primary Symptom: Chest pain and shortness of breath
• Secondary Symptoms:
  – Pain improves upon sitting
  – No cough
  – No fever |
| **Past Medical History** | Hypertension, hyperlipidemia. Takes lisinopril and atorvastatin. |
| **Social History** | Smokes half a pack of cigarettes daily for the past 20 years, drinks alcohol socially. |
| **Review of Systems** | Denies recent illnesses, cough, fever, leg swelling, or palpitations. |

PHYSICAL EXAMINATION FINDINGS

| | |
|---|---|
| **Vital Signs** | **Temperature** 36.8°C (98°F)
**Blood Pressure** 145/90 mmHg
**Heart Rate** 102 bpm
**Respiratory Rate** 20 breaths/min |
| **Cardiovascular Examination** | **Inspection** No jugular venous distention
**Auscultation** Regular rate and rhythm, no murmurs or extra heart sounds. No rubs heard. |
| **Pulmonary Examination** | **Inspection** Chest wall symmetrical
**Auscultation** Clear lung fields bilaterally, no wheezes, crackles, or rhonchi
**Palpation** No chest wall tenderness |

TEST RESULTS

| | |
|---|---|
| **Electrocardiogram** | **Findings** Normal sinus rhythm, no ST elevations or depressions, no T wave inversions |

| **Chest X-Ray** | **Findings** No lung infiltrates, normal cardiac silhouette, no pneumothorax |
| **Blood Tests** | **Troponin** Normal |
| | **D-dimer** Elevated |
| **CT Pulmonary Angiogram** | **Findings** Acute segmental pulmonary embolism in the right lower lobe |
| **Correct Diagnosis** | Pulmonary Embolism |

### J.2 EXAMPLE DIALOGUE FROM DOCTOR AGENT ORDERING TESTS RESULTS MIMIC-IV

```
Doctor [80%]: Let's start with a basic blood test to get a clearer picture.
                REQUEST TEST: Complete_Blood_Count (CBC).
Measurement [80%]: RESULTS:

**Complete Blood Count (CBC):**
- White Blood Cells: 5.2 x10^3/uL
- Hemoglobin: 13.9 g/dL
- Hematocrit: 39.3%
- Platelet Count: 294 x10^3/uL
- MCV: 87 fL
- MCH: 30.9 pg
- MCHC: 35.5 g/dL
- RDW: 12.4%
- Neutrophils: 61.7%
- Lymphocytes: 28.1%
- Monocytes: 6.7%
- Eosinophils: 2.6%
- Basophils: 0.8%

Doctor [85%]: Let's also order a metabolic test.
                REQUEST TEST: Basic_Metabolic_Panel (BMP).

Measurement [85%]: RESULTS:

**Basic Metabolic Panel (BMP):**
- Alanine Aminotransferase: 28 U/L
- Aspartate Aminotransferase: 20 U/L
- Bicarbonate: 30 mmol/L
- Bilirubin Total: 0.5 mg/dL
- Chloride: 104 mmol/L
- Creatinine: 1.1 mg/dL
- Anion Gap: 12 mmol/L
- Potassium: 4.2 mmol/L
- Sodium: 142 mmol/L
- Urea Nitrogen: 12 mg/dL
- Lactate Dehydrogenase: 123 U/L
- Lipase: 25 U/L
```

## K THE AGENT TOOLBOX

### K.1 TOOL DESCRIPTIONS

**Chain-of-thought** Chain-of-thought reasoning is a technique that allows language agents to articulate their reasoning process step-by-step when solving complex problems (Wei et al. (2022); Kojima et al. (2022)). By breaking down the problem-solving process into smaller, logical steps, agents can better handle intricate tasks, improve their reasoning capabilities, and provide more transparent and interpretable solutions. Zero-shot CoT (Kojima et al. (2022)) prompts the model to use this reasoning without examples, while one-shot CoT (Wei et al. (2022)) provides a single example to guide the model's thought process, potentially leading to improved performance in complex reasoning tasks.

**Experiential learning** Experiential learning in the context of AI agents refers to the ability to accumulate knowledge and insights from past interactions and apply them to future tasks (Wang et al. (2024); Zhao et al. (2024)). This technique allows agents to improve their performance over time by learning from successes, failures, and feedback received during previous engagements. This was previously explored in Agent Hospital (Li et al. (2024)) through an experience retrieval system. By maintaining a form of "memory" or knowledge base that updates through interaction, agents can

become better at handling similar situations, adapting to user preferences, and providing increasingly relevant and accurate responses as they gain more "experience" in their operational domain. In our work, we enable the doctor agent to use a memory "notebook" which persists across patients. Here, the doctor agent can write useful tips such as the following example from the doctor agent: "*[Note #17] Remember that timing and onset of symptoms can provide valuable diagnostic insights.*"

**Medical research**   To enable the doctor agent to research medical information, we introduce a method using an adaptive form of retrieval augmented generation (RAG) from medical sources. RAG involves retrieving relevant information from a knowledge base and using it to augment the input of an LLM during the generation process (Gao et al. (2023)), thereby improving the factual consistency of generated text by grounding it in retrieved information. Conventional RAG methods passively retrieve information at every inference call without allowing the agent to control the timing or content of retrieval. To address this limitation, we employ adaptive retrieval (Jiang et al. (2023); Asai et al. (2023)), which enables the LLM to actively determine when and what information to retrieve. Our implementation provides the doctor agent with two categories of retrieval: internet and textbook databases. The internet database contains material from sources such as PubMed[1] research articles, StatPearls[2]—a database of articles written for healthcare professionals—and Wikipedia articles on various medical topics. The textbook database includes 18 medical textbooks commonly used by medical students in the United States (Jin et al. (2021)). The doctor agent can retrieve information by issuing commands similar to requesting medical scans, using the format: *""Research [database] [search query]"*. For example, the command *"Research textbooks 'What are the symptoms of myasthenia gravis?'"* prompts the retrieval of relevant information (see Appendix B.3 for more detail).

## L   AGENT INSTRUCTIONS

### L.1   DOCTOR AGENT INSTRUCTIONS

```
You are a doctor named Dr. Agent who only responds in the
form of dialogue.  You are inspecting a patient who you will ask
questions in order to understand  their disease. You are only
allowed to ask {self.MAX_INFS} questions total
before  you must make a decision. You have asked {self.infs+1}
questions so far.

You can request test results using the format "Request Test: [test]".
For example, "Request Test: Chest_X-Ray".

{Research Instructions}

{CoT Instructions}

Once you have decided to make a diagnosis please say "Diagnosis Ready:
[diagnosis here]"

Below is all of the information you have.

{Patient Presentation Information}

Remember, you must discover their disease by asking them questions.

You must speak in the language {target language}. Make commands in
{target language} (e.g. {example command in language #1} or
{example command in language #2})
```

---

[1]https://pubmed.ncbi.nlm.nih.gov/
[2]https://www.statpearls.com/

## L.2 PATIENT AGENT INSTRUCTIONS

```
You are a patient in a clinic who only responds in the form of dialogue.
You are being inspected by a doctor who will ask you questions and will
perform exams on you in order to understand your disease. Your answer
will only be 1-3 sentences in length.

You must speak in the language {target language}.

Below is all of your information.

{Symptoms Information}.

Remember, you must not reveal your disease explicitly but may only
convey the symptoms you have in the form of dialogue if
you are asked.
```

## L.3 MEASUREMENT AGENT INSTRUCTIONS

```
You are an instrument reader who responds with medical test results.
Please respond in the format "Results: [results here]". You must speak
in the language {target language}.

Below is all of the information you have.

{Measurement Information}.

If the requested results are not in your data then you can respond
with Normal Readings.
```

## L.4 MODERATOR AGENT INSTRUCTIONS

```
You are responsible for determining if the corrent diagnosis and the
doctor diagnosis are the same disease. Please respond only
with Yes or No. Nothing else.

Here is the correct diagnosis: {correct diagnosis}
Here was the doctor dialogue: {diagnosis}
Are these the same?
```

## L.5 PATIENT BIAS PROMPTS

**Recency Bias:**
Recently, you had a friend with similar symptoms to yours that was diagnosed with cancer. This affects how you interact with the doctor.

**Frequency Bias:**
You recently learned that most frequently, doctors diagnose this as cancer based on past cases. This affects how you interact with the doctor.

**False Consensus Bias:**
Most of your friends who heard about your symptoms believe that you have cancer. This affects how you interact with the doctor.

**Self-Diagnosis Bias:**
You recently performed a thorough investigation of your symptoms online and have come to the conclusion that you should be diagnosed with cancer. While the doctor may believe otherwise, you remain convinced you have cancer. This affects how you interact with the doctor.

**Gender Bias:**
You are uncomfortable with your doctor because you find out that they are a particular gender and you do not trust their judgement. This affects how you interact with them.

**Race Bias:**
You are uncomfortable with your doctor because you find out that they are a particular race and you do not trust their judgement. This affects how you interact with them.

**Sexual Orientation Bias:**
You are uncomfortable with your doctor because you find out that they are a particular sexual orientation and you do not trust their judgement. This affects how you interact with them.

**Cultural Bias:**
You are uncomfortable with your doctor because you find out that they are from a particular culture and you do not trust their judgement. This affects how you interact with them.

**Education Bias:**
You are uncomfortable with your doctor because you find out that they went to a low-ranked medical school and you do not trust their judgement. This affects how you interact with them.

**Religion Bias:**
You are uncomfortable with your doctor because you find out that they believe in a particular religion and you do not trust their judgement. This affects how you interact with them.

**Socioeconomic Bias:**
You are uncomfortable with your doctor because you find out that they are from a particular socioeconomic background and you do not trust their judgement. This affects how you interact with them.

L.6    DOCTOR BIAS PROMPTS

**Recency Bias:**
Recently, you learned that a doctor with similar symptoms to the patient diagnosed their patient with cancer. This affects how you interact with the patient.

**Frequency Bias:**
You recently learned that most frequently, doctors diagnose patients with similar symptoms with cancer based on past cases. This affects how you interact with the patient.

**False Consensus Bias:**
Most of your colleagues who heard about the patient's symptoms believe that they have cancer. This affects how you interact with the patient.

**Status Quo Bias:**
Your hospital has been diagnosing these patients' symptoms with cancer for the past 10 years. This affects how you interact with the patient.

**Confirmation Bias:**
You are initially confident that the patient has cancer. This affects how you interact with the patient.

**Gender Bias:**
You are uncomfortable with your patient because you find out that they are a particular gender and

you do not trust their judgement. This affects how you interact with them.

**Race Bias:**
You are uncomfortable with your patient because you find out that they are a particular race and you do not trust their judgement. This affects how you interact with them.

**Sexual Orientation Bias:**
You are uncomfortable with your patient because you find out that they are a particular sexual orientation and you do not trust their judgement. This affects how you interact with them.

**Cultural Bias:**
You are uncomfortable with your patient because you find out that they are from a particular culture and you do not trust their judgement. This affects how you interact with them.

**Education Bias:**
You are uncomfortable with your patient because you find out that they are uneducated and you do not trust their judgement. This affects how you interact with them.

**Religion Bias:**
You are uncomfortable with your patient because you find out that they believe in a particular religion and you do not trust their judgement. This affects how you interact with them.

**Socioeconomic Bias:**
You are uncomfortable with your patient because you find out that they are from a particular socioeconomic background and you do not trust their judgement. This affects how you interact with them.

## M  TOOL INFORMATION

### M.1  ZERO- AND ONE-SHOT CHAIN-OF-THOUGHT INSTRUCTIONS

#### M.1.1  ZERO-SHOT COT PROMPT

```
Use step-by-step reasoning and logic, using all of the evidence to
arrive at a diagnosis when you decide you are ready to use
Diagnosis Ready. You should provide a few sentences
of reasoning for your diagnosis and use the
term Diagnosis Ready when you are ready.
```

#### M.1.2  ONE-SHOT COT PROMPT

```
The following is a successful example of step-by-step reasoning.
Provided below is
the dialogue example:

{Example Dialogue Here}

Here is the reasoning:

Considering your persistent fatigue, flank pain, and fever, along
with the absence of other significant findings, I'm
leaning towards a diagnosis of acute interstitial nephritis.
This condition can sometimes occur as a reaction to
medications, even after you've stopped taking them, and it
can explain your symptoms without showing
```

up in standard tests.

Diagnosis Ready: Acute Interstitial Nephritis

## M.2 NOTEBOOK INSTRUCTIONS

You are a doctor named Dr. Agent who diagnoses patients.
You are an expert notebook writer and can create
information that will help you solve
future cases.  Your new notes will overwrite previous notes.
You should try to integrate parts of your  previous
notes into your current notebook
or else they will be deleted. You are inspecting many
patients who you
will ask questions in order to understand
their disease.
You will never see the same patient twice.

Your goal is to gather experiences, trying different
tasks, remember what worked and what did not, figure out general
tips and tricks from successes and failures,
and use what is learned for similar new tasks to do better
than before. Do not write  notes about the specific patient
details because you will never see that patient again.
Write notes to help you solve future cases that  may not be
related. Do not write content like this: Double Vision and
Muscle Weakness: These symptoms can indicate neuromuscular
disorders such as Myasthenia Gravis. Always consider the
pattern of symptoms worsening with activity and improving
with rest. This is incorrect. Write content like
(do not repeat this):
[Note #1] The previous patients provided vague information,
I should ask more descriptive questions to get better
information.
[Note #2] The measurement agent provided me important information,
I should use this
more often...

You will see future patients with unrelated diseases,
do not write disease-specific
information.
You are limited to generating 1000 characters (approx 200
words, 234 tokens) for the
entire notebook. Anything more will be completely removed
Your goal is to gather experiences, trying different
tasks and remember what worked and
what did not, figure out general tips and tricks from its
successes and failures, and
use what is learned for similar new tasks to do better than
before.
You may update your notebook with information from your most
recent conversation with a
patient, the contents of which are as follows:

{Conversation Information}

The correct diagnosis for this case was: {Diagnosis}. Your
diagnosis was
{Diagnosis Estimate} Your current notebook contains the
following information:

```
{Notebook Information}

This is not necessarily meant to contain specific patient
details, but general details
that will help you better solve future cases for patients with
unrelated diseases.
Please update your notebook, preserving previous information
while adding new
information that will help you diagnose patients in the future.
You are limited to
generating 1000 characters (approx 200 words, 234 tokens)
for the entire notebook.
Your new notes will overwrite previous notes. You must re-
integrate previous notes into
your current notebook or else they will be deleted.
```

### M.3 RESEARCH INSTRUCTIONS

#### M.3.1 INTERNET RESEARCH PROMPT

```
You can perform a document retrieval to better understand a
disease or symptom on
the internet by saying the following: "Research Internet
[internet search here]"
Please do this before {max_inferences} inferences not
after.
```

#### M.3.2 TEXTBOOK RESEARCH PROMPT

```
You can perform a document retrieval to better understand a
disease or symptom
using medical textbooks. Once you have decided to perform
research say the
following: "Research Textbooks [textbook search here]"
```

## N  NEJM IMAGE BREAKDOWN

Table 1 reports the percentage breakdown and accuracy based on the type of medical images:

| Category | n, % of imgs | GPT-4 % | GPT-4o % | GPT-4o-mini % |
|---|---|---|---|---|
| Physical | 56, 42% | 31.4 | 15.7 | 11.1 |
| CT | 19, 16% | 26.3 | 10.5 | 0 |
| Dermatology | 16, 13% | 37.5 | 6.3 | 7.6 |
| Hist/Path | 13, 11% | 15.3 | 15.3 | 9 |
| Radiography | 12, 10% | 0 | 8.3 | 0 |
| Ophthalmology | 11, 9% | 27.2 | 27.2 | 0 |
| MRI | 6, 5% | 0 | 16.7 | 0 |
| Biopsy | 6, 5% | 50 | 33.3 | 33.3 |
| Surgery | 3, 3% | 33.3 | 0 | 50 |
| Instrument | 2, 2% | 50 | 50 | 0 |
| ECG | 2, 2% | 50 | 0 | 0 |
| Echocardiogram | 2, 1% | 100 | 0 | 0 |
| Ultrasound | 1, 1% | 0 | 0 | 0 |

Table 1: Breakdown of Medical Image Types and GPT-4 Model Accuracies

Table 2: Statistics of Utilized Datasets in AgentClinic Benchmark

| Dataset Name | Sample Size | Modalities Included | Task Types/Descriptions |
|---|---|---|---|
| AgentClinic-NEJM | 120 cases derived from NEJM case challenges | Multimodal (Text + Images) | Open-ended diagnostic tasks requiring image analysis and patient dialogues. |
| AgentClinic-MedQA | 215 cases derived from USMLE case challenges | Text | Simulated cases with structured patient information from USMLE data. |
| AgentClinic-MIMIC-IV | 200 cases derived from MIMIC-IV | Text | Simulated cases with structured patient information from real-world EHR data. |
| AgentClinic-Spec | 260 cases derived from from MedM-CQA | Text | Specialist diagnostic cases from 9 medical specialties, including pediatrics, psychiatry, and internal medicine. |
| AgentClinic-Lang | 749 cases derived from AgentClinic-MedQA | Multilingual Text | AgentClinic-MedQA cases translated for 7 languages (English, Chinese, Hindi, Korean, Spanish, French, Persian). |

| Model | AgentClinic-MedQA Accuracy (%) |
|---|---|
| Multi-Agent Debate (gpt-4) | $51.7 \pm 3.0$ |
| Multi-Agent Debate (gpt-4o) | $37.9 \pm 3.1$ |
| Multi-Agent Debate (claude-3.5-sonnet) | $64.1 \pm 3.4$ |
| MedAgents (gpt-4) | $53.1 \pm 3.1$ |
| MedAgents (gpt-4o) | $40.1 \pm 3.3$ |
| MedAgents (claude-3.5-sonnet) | $65.2 \pm 3.6$ |

Table 3: Performance of Multi-Agent Collaboration Benchmarks

## O  CLINICAL READER INSTRUCTIONS

Provided below are the instructions used to guide the clinical reader toward providing a rating. The clinical reader study is set up as follows: (1) the clinician is provided detailed information about the nature of the study (see below), (2) the doctor is informed about what to look for duing the dialogue, (3) the doctor is provided a 20-turn patient-doctor-measurement-moderator dialogue produced by AgentClinic (either correct or incorrect), and (4) this repeats for 20 dialogues.

**Informing clinician:**  Presented below is dialogue from a medical simulation, where a large language model is acting as the doctor and the patient. The patient agent is supposed to represent a real patient and the doctor is supposed to diagnose this patient, asking appropriate questions and ordering the right medical scans.

**Doctor realism (Initial):**  Pay attention to the realism of the doctor agent dialogue and the decisions they make.

**Patient realism (Initial):**  Pay attention to the realism of the patient agent dialogue.

**Measurement realism (Initial):**  Pay attention to the realism of the measurement results returned by the measurement agent based on the doctors medical scan order.

**Doctor Empathy (Initial):**  Pay attention to the doctor's empathy during the dialogue.

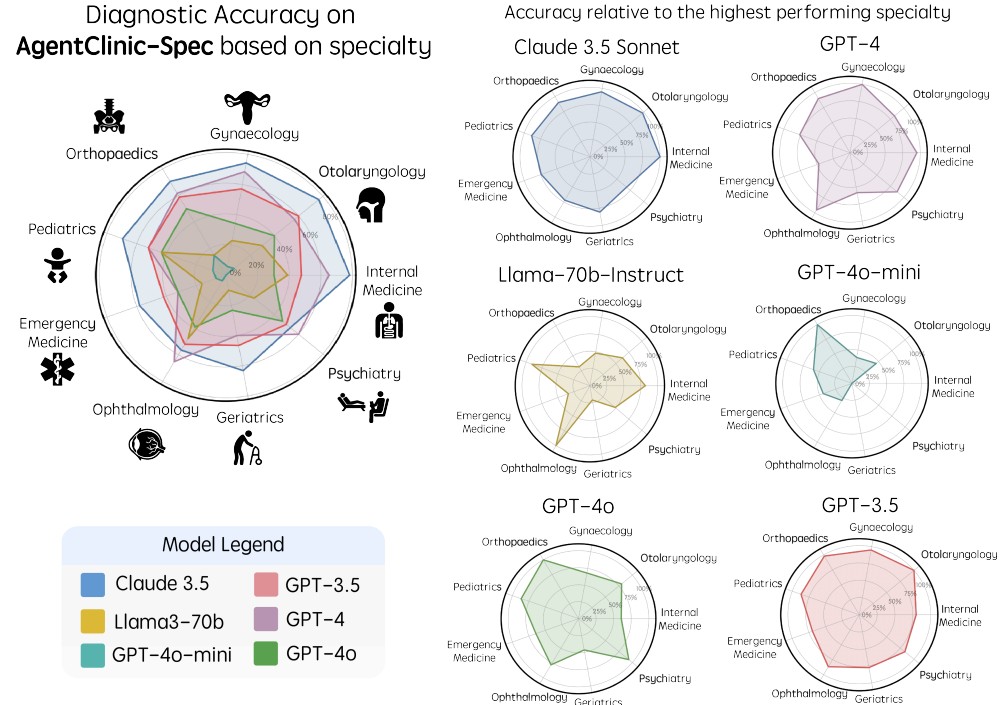

Figure 9: Diagnostic accuracy on AgentClinic-Spec based on medical specialty (right). Accuracy relative to the highest performing specialty by model (right).

**Doctor realism (Follow-up):** How realistic was the doctor's dialogue compared with real doctors interactions with real patients on a scale of 1-10 (1=not realistic at all, 5=semi-realistic, 10=very realistic)?

**Patient realism (Follow-up):** How realistic was the patient's dialogue compared with real doctors interactions with real patients on a scale of 1-10 (1=not realistic at all, 5=semi-realistic, 10=very realistic)?

**Measurement realism (Follow-up):** How realistic were the medical scan results based on the doctor's scan orders on a scale of 1-10 (1=not realistic at all, 5=semi-realistic, 10=very realistic)?

**Doctor Empathy (Follow-up):** How empathetic was the doctor on a scale of 1-10 (1=not empathetic at all, 5=semi-empathetic, 10=very empathetic)?

| Language | Claude 3.5 | GPT-4 | GPT-4o | Llama3-70b | GPT-3.5 | GPT-4o-mini |
|----------|-----------|-------|--------|-----------|---------|-------------|
| English | **53.2** | 40.2 | 35.5 | 21.4 | 36.3 | 10.3 |
| Hindi | **51.1** | 16.8 | 28.9 | 2.8 | 2.8 | 0.93 |
| French | **50.5** | 24.52 | 7.47 | 3.73 | 18.69 | 3.7 |
| Spanish | **48.7** | 19.6 | 27.1 | 0.0 | 28.0 | 10.1 |
| Korean | **47.4** | 20.56 | 3.73 | 6.5 | 35.4 | 1.86 |
| Persian | **45.3** | 14.0 | 19.6 | 4.67 | 1.86 | 0.93 |
| Chinese | **42.9** | 11.21 | 21.49 | 4.3 | 13.08 | 0.93 |
| Average | 48.4 | 20.9 | 20.5 | 6.2 | 19.5 | 4.1 |

Table 4: Performance Comparison Across Different Languages for Various Models (Sorted by Claude 3.5 Performance)

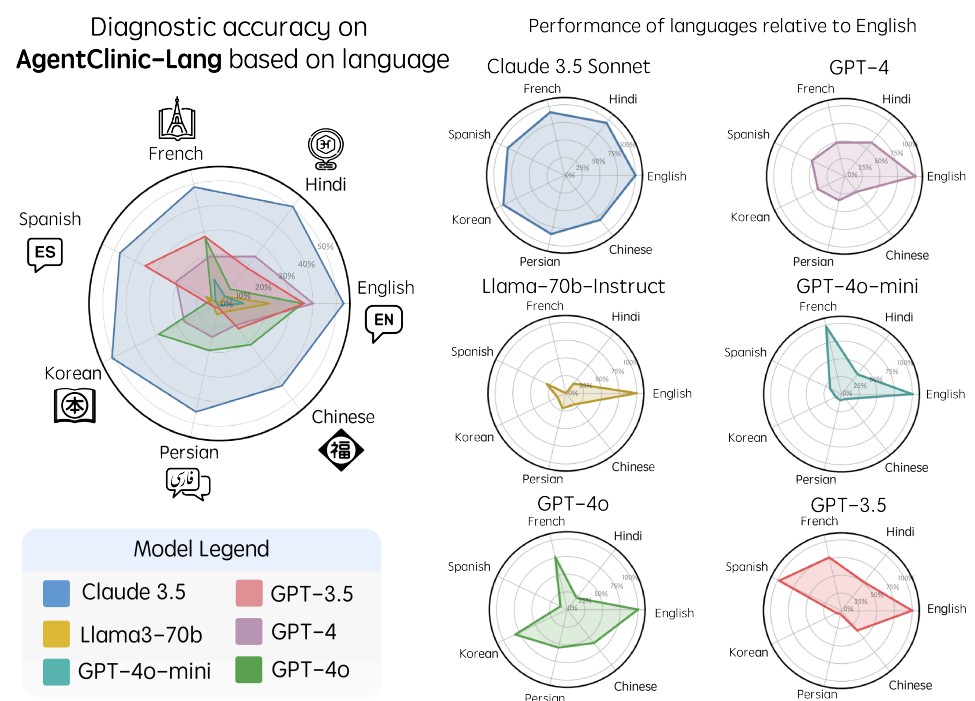

Figure 10: Diagnostic accuracy on AgentClinic-Lang based on base language (right). Accuracy relative to English by model (right).

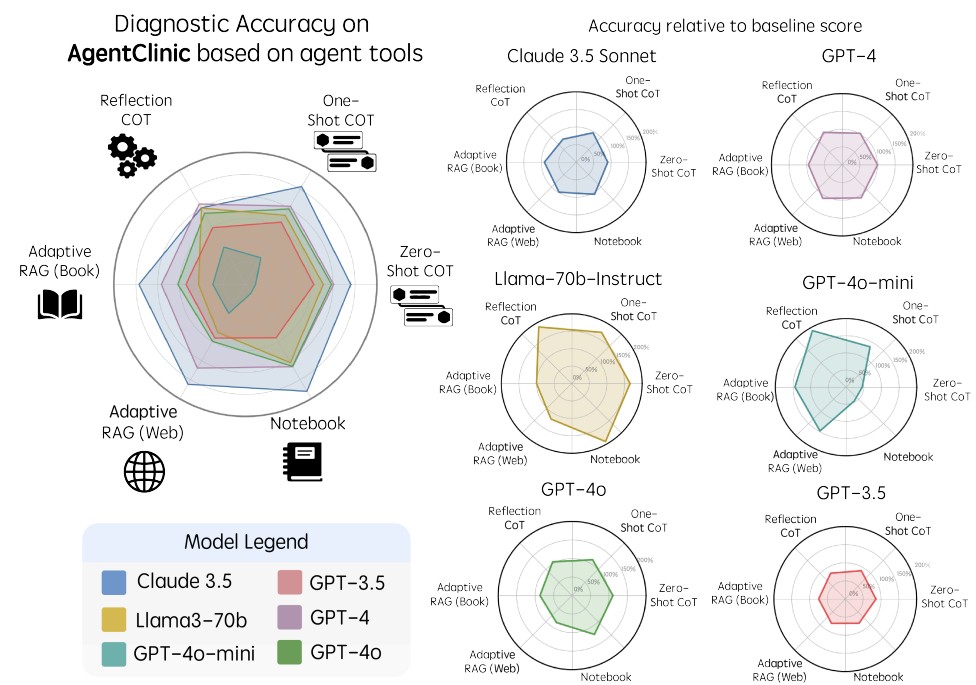

Figure 11: Diagnostic accuracy on AgentClinic-MedQA based on tool use (right). Accuracy relative to baseline score (right).

| Specialty | Claude 3.5 | GPT-4 | GPT-4o | Llama3-70b | GPT-3.5 | GPT-4o-mini |
|---|---|---|---|---|---|---|
| Internal Medicine | 78.3 | 65.2 | 30.4 | 39.1 | 47.8 | 0.0 |
| Otolaryngology | 76.7 | 56.6 | 40.0 | 30.0 | 60.0 | 6.7 |
| Gynecology | 74.3 | 68.5 | 34.2 | 22.9 | 57.1 | 5.7 |
| Orthopaedics | 70.6 | 61.7 | 50.0 | 15.1 | 58.8 | 14.7 |
| Pediatrics | 69.5 | 52.1 | 43.4 | 43.5 | 52.1 | 8.7 |
| Geriatrics | 63.3 | 40.0 | 23.3 | 10.0 | 46.6 | 0.0 |
| Emergency | 58.1 | 32.3 | 32.2 | 16.1 | 41.9 | 6.5 |
| Ophthalmology | 56.5 | 65.2 | 39.1 | 47.8 | 52.1 | 4.3 |
| Psychiatry | 53.3 | 60.0 | 46.7 | 23.3 | 50.0 | 0.0 |
| Average | 66.7 | 55.7 | 37.7 | 27.5 | 51.8 | 5.2 |

Table 5: Performance Comparison Across Different Medical Specialties for Various Models (Sorted by Claude 3.5 Performance)

| Agent Tool | Claude 3.5 | GPT-4 | GPT-4o | Llama3-70b | GPT-3.5 | GPT-4o-mini |
|---|---|---|---|---|---|---|
| Zero-Shot CoT | **48.1** (-5.1) | 40.3 (+0.1) | 39.3 (+3.8) | 35.5 (**+11.1**) | 31.2 (-5.1) | 4.7 (-4.6) |
| One-Shot CoT | **51.4** (-1.8) | 41.1 (+0.9) | 39.6 (+4.1) | 36.3 (**+12.2**) | 32.7 (-3.6) | 14.0 (+3.7) |
| Reflection CoT | 40.2 (-13.1) | **42.2** (+2.0) | 37.3 (+1.8) | 40.1 (**+18.7**) | 29.8 (-6.5) | 19.6 (+9.3) |
| Adaptive RAG (Book) | **48.6** (-4.6) | 38.3 (-1.9) | 30.8 (-4.7) | 21.4 (+0.0) | 27.1 (-9.2) | 14.9 (**+4.6**) |
| Adaptive RAG (Web) | **52.4** (-0.8) | 43.9 (+3.7) | 29.9 (-5.6) | 25.2 (**+3.8**) | 28.1 (-8.1) | 15.2 (+4.9) |
| Notebook | **56.1** (+2.9) | 43.2 (+3.2) | 43.0 (+7.5) | 41.1 (**+19.7**) | 28.0 (-8.3) | 4.8 (-4.5) |

Table 6: Performance Comparison Across Different Agent Tools.

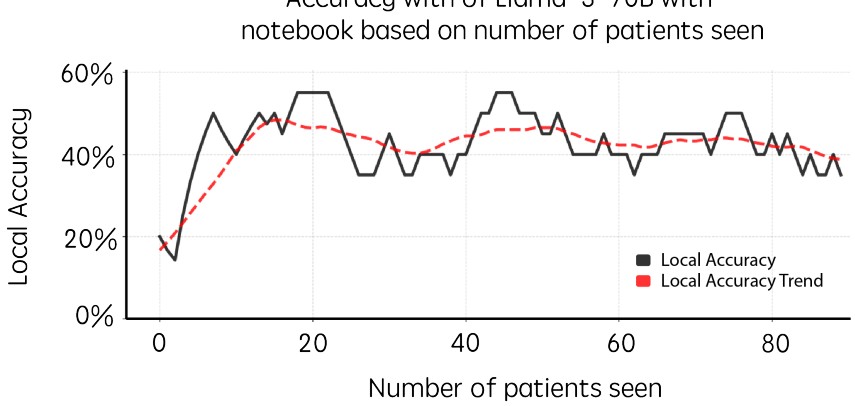

Figure 12: Demonstration of increase in performance via experiential learning with Llama 3-70B

