# OpenReview forum: "AgentClinic: a multimodal agent benchmark to evaluate AI in simulated clinical environments"
_ICLR.cc/2025/Conference — Submitted to ICLR 2025_

### Official Review · Reviewer_QGPA · 2024-11-03

**Soundness:** 3
**Presentation:** 2
**Contribution:** 2
**Rating:** 6
**Confidence:** 4

**Summary:**

In this work, the authors introduce AgentClinic, a multimodal benchmark for evaluating LLMs within simulated clinical environments. The pipeline’s performance is assessed using the MedQA, MIMIC-IV, and NEJM datasets.

**Strengths:**

Some conclusions may hold clinical relevance

**Weaknesses:**

My primary comments are as follows:

1. The paper is challenging to follow, as many crucial details are buried in the lengthy appendix rather than presented in the main text. For instance, it is unclear what specific biases are explored in this work and how they are defined. Additionally, how agent tools are integrated within the pipeline, what specific data are extracted from the MIMIC-IV dataset, the nature of the NEJM case challenge dataset, and whether it includes QA pairs all need clarification. Furthermore, the difference between specialist case use reports and general QA tasks should be explained. I recommend providing brief explanations for any non-standard terms in the main text for improved clarity.

2. Although the paper claims to simulate a real-world clinical environment, several settings and use cases seem impractical. For example, what is the intended purpose of the measurement agent? Why can’t the complete records be provided directly, or why doesn’t the doctor agent extract values directly from the database? Additionally, evaluating a patient agent seems unrealistic within actual clinical workflows.

3. Since the study focuses on evaluating LLMs within clinical contexts, the benchmark appears to lack some medical LLMs.

**Questions:**

Please refer to the weaknesses.

---

> ### Author Response · Authors · 2024-11-24
>
> We appreciate the reviewer’s recognition of our work’s clinical relevance and for the helpful feedback. We provide a point-by-point response below:
>
> > 1a. Many crucial details are in the appendix rather than presented in the main text. For instance, it is unclear what specific biases are explored in this work and how they are defined.
>
> Thank you for this observation. We recognize the importance of including critical details in the main text for clarity and accessibility. We had many results in this work and had limited space in the manuscript for extensive details. However, according to your advice, we have incorporated a succinct summary of the key biases explored in this work into the **main text**. This includes an overview of the two categories of biases (cognitive and implicit), their definitions, and their specific clinical relevance as is now presented in Section 2 *Language agent biases*, which incorporates relevant details outlined in Appendix A.2, E, E.1-E.2, and L.5. These updates aim to provide a more self-contained narrative in the main body.
>
> > 1b. Additionally, how agent tools are integrated within the pipeline, what specific data are extracted from the MIMIC-IV dataset, the nature of the NEJM case challenge dataset, and whether it includes QA pairs all need clarification.
>
> We thank you for the suggestion. To clarify, agent tools are integrated into the pipeline as modular components that the doctor agent invokes for specific tasks, such as retrieving test results, searching references, or summarizing notes. For MIMIC-IV cases, the extracted data includes de-identified patient histories, lab results, and diagnostic findings, focusing on structured data relevant to the presented symptoms. The NEJM case challenge dataset contains diagnostic questions derived from real clinical cases and includes QA pairs for validation. To further clarify this, we provide detailed set of replication scripts for cases in our work are provided at the following de-identified link: https://github.com/AgentClinic/AgentClinicAnonymized/tree/main/generate_cases.
>
> > 1c. Furthermore, the difference between specialist case use reports and general QA tasks should be explained. I recommend providing brief explanations for any non-standard terms in the main text for improved clarity.
>
> To better clarify the distinction, we have included the following explanation in the main text: *"Specialist cases use reports that are derived from datasets focusing on specific medical specialties (e.g., internal medicine, psychiatry) and are designed to simulate complex diagnostic scenarios requiring in-depth expertise. In contrast, general QA tasks involve static, single-turn multiple-choice questions such as those found in medical licensing exams."* This has been added to Section 3.4: Specialist and Multilingual Cases in the Results section. We believe that this better clarifies the difference between specialist case use reports and general QA tasks and better contextualizes the specialist cases.
>
> > 2a. Although the paper claims to simulate a real-world clinical environment, several settings and use cases seem impractical. For example, what is the intended purpose of the measurement agent?
>
> The inclusion of the measurement agent is intended to simulate a realistic clinical workflow where a doctor requests specific imaging or laboratory tests rather than having unrestricted access to all retrospectively collected data. This design choice aims to better reflect real-world constraints in clinical environments. The distinction between an agent and a tool lies in their interaction style: tools require highly specific and structured instructions, while the measurement agent offers a flexible interface capable of interpreting diverse linguistic styles used by different LLMs. This flexibility enables smoother integration across models that may vary in their use of acronyms, word order, or phrasing. While the decision to use a measurement agent versus a database agent or a simple tool is ultimately somewhat arbitrary, both approaches can serve the purpose of enabling the doctor agent to collect necessary data. Discussions with our collaborating clinicians further validated the relevance of the measurement agent, as it aligns with emerging systems (e.g., conversational copilots) designed to retrieve specific patient information from electronic health records (EHRs) through interactive queries.

---

> > ### Author Response · Authors · 2024-11-24
> >
> > > 2b. Why can’t the complete records be provided directly, or why doesn’t the doctor agent extract values directly from the database?
> >
> > In clinical practice, complete records are not typically provided upfront to clinicians. Instead, medical workflows are inherently sequential decision-making processes conducted under conditions of incomplete information. These workflows involve iterative data collection, where clinicians weigh various trade-offs, such as urgency, safety, cost, and invasiveness, to determine the next appropriate steps. For example, decisions about whether to proceed with a simple blood panel or an expensive and time-consuming MRI often depend on the information already gathered and the clinical context at hand. Providing the doctor agent with direct access to complete records would bypass these critical aspects of real-world clinical decision-making. The sequential approach better simulates the nuanced processes clinicians follow in actual practice, moving beyond the static "full case to question" format often found in medical QA problems.
> >
> > > 3c. Additionally, evaluating a patient agent seems unrealistic within actual clinical workflows.
> >
> > The inclusion of a patient agent serves as a mechanism to more effectively evaluate an LLM's ability to perform clinical tasks, whether in supporting a doctor or simulating a doctor in scenarios where a specialist is unavailable (e.g., in remote or underserved settings). The patient agent allows for the simulation of realistic clinical scenarios in which patients may not understand medical terminology, lack the ability to interpret their symptoms and test results, and are unsure of what information to provide or when to provide it. This places greater emphasis on the anamnesis skills of the doctor language agent, mimicking real-world clinical interactions where eliciting relevant information from patients is a key challenge.
> >
> > In contrast, conventional QA settings, where complete cases are pre-assembled and presented to the model, remove this interactive component. Such setups presuppose that all pertinent information has already been provided, often reflecting idealized scenarios rather than the complexities of actual clinical workflows.
> >
> > Furthermore, the patient agent enables the introduction of novel, patient-centered evaluation metrics, such as the empathy demonstrated by the doctor agent, the patient’s likelihood to comply with the proposed diagnosis and treatment, and the willingness to revisit or recommend the doctor. These metrics offer a more comprehensive assessment of clinical performance, focusing not only on diagnostic accuracy but also on factors critical to effective healthcare delivery.
> >
> >
> > > 3. Since the study focuses on evaluating LLMs within clinical contexts, the benchmark appears to lack some medical LLMs.
> >
> > We agree that the inclusion of results from medical LLMs would strengthen this work. In response to this feedback, we have incorporated evaluations of four additional SOTA medical LLMs: MedLlama3-8B, Meditron-70B, PMC-Llama-7B, and OpenBioLLM-70B. We have also updated Figures 2 and 3 to reflect these changes. The results for these models on the AgentClinic-MedQA and AgentClinic-MIMIC-IV benchmarks are presented below:
> >
> > | **Benchmark**            | **Model**           | **Accuracy (% ± CI)**  |
> > |---------------------------|---------------------|-------------------------|
> > | **AgentClinic-MedQA**    | MedLlama3-8B       | 31.4 ± 2.9             |
> > |                           | Meditron-70B       | 29.1 ± 2.4             |
> > |                           | PMC-Llama-7B       | 23.6 ± 2.1             |
> > |                           | OpenBioLLM-70B     | 58.3 ± 4.2             |
> > | **AgentClinic-MIMIC-IV** | MedLlama3-8B       | 29.7 ± 2.6             |
> > |                           | Meditron-70B       | 25.5 ± 2.3             |
> > |                           | PMC-Llama-7B       | 34.3 ± 3.0             |
> > |                           | OpenBioLLM-70B     | 38.1 ± 3.2             |
> >
> > We have incorporated these results into our updated manuscript and hope that you find this improves the quality of our work.

---

> > > ### Comment · Reviewer_QGPA · 2024-11-27
> > > **Response to author rebuttal**
> > >
> > > Thanks the authors for their response. The added details improve the readability of the paper. I've updated my score. But the measurement agents are still questionable to me. I don't see the advantage of using an extra agent to retrieve results rather than using sql or directly providing the results in the risk of adding extra complexities and hallucinations, especially the authors find there are errors in the retrieved results.

---

> > > > ### Author Response · Authors · 2024-11-27
> > > >
> > > > Thank you for the detailed feedback. Regarding the measurement agent, its purpose is to simulate realistic clinical workflows where a doctor must request specific tests rather than having unrestricted access to all patient data. The measurement agent provides a *flexible* interface to interpret diverse inputs across models, accommodating differences in phrasing or terminology, much like a doctor requesting a nurse to perform blood work or an x-ray and returning the results. Previous studies indicate that less flexible string matching with LLMs disproportiantely advantages models which can fit the *exact query* required for the problem, where it is demonstrated that "nonlinear or discontinuous metrics produce apparent emergent abilities, whereas linear or continuous metrics produce smooth, continuous, predictable changes in model performance" [1]. From these results, we believe that, while an SQL or DB agent would be a possible tool to incorporate instead of a measurement/SQL/DB agent, it would prevent models from extracting the proper information when there are small variations of the request leading to the "emergent" performance curves demonstrated in [1] instead of the smooth continuous performance curves shown in our work and in others. However, we also recognize that integrating the measurement agent could also introduce complexities, and we appreciate your point about potential risks of errors or hallucinations. These trade-offs, as well as the potential for using a DB/SQL tool, are now acknowledged in the revised discussion, and would be interesting to consider in future studies.
> > > >
> > > > [1] Schaeffer, Rylan, Brando Miranda, and Sanmi Koyejo. "Are emergent abilities of large language models a mirage?." Advances in Neural Information Processing Systems 36 (2024).

---

> ### Comment · Reviewer_QGPA · 2024-11-28
> **Response to author rebuttal**
>
> Thanks to the authors for their response. That's a fair point. Though in my opinion the settings in this work may still have gaps from the real practice, I'll not object to accepting this work.

---

### Official Review · Reviewer_6dqE · 2024-11-04

**Soundness:** 3
**Presentation:** 3
**Contribution:** 3
**Rating:** 8
**Confidence:** 4

**Summary:**

The article establishes a dynamic evaluation benchmark by simulating doctor-patient dialogues to assess model capabilities. Detailed experiments were conducted in scenarios involving bias, multilingualism, multiple departments, and multimodality, providing the community with a comprehensive evaluation framework.

**Strengths:**

- Built an evaluation framework for multiple agents to assess the performance of LLM models from various dimensions, including diagnostic performance and patient experience.
- Constructed a comprehensive test set to measure LLM's diagnostic capabilities in a simulated environment from multiple settings, including performance of different models in biased environments, multilingual, multi-department, and multi-modal scenarios.
- Tested several representative closed-source and open-source models and provided an objective evaluation.

**Weaknesses:**

- The article's simulation of interactions between multiple agents is relatively simple, with the main interactions in the experiment limited to between the doctor agent and the patient agent.
- The article does not conduct an in-depth analysis of the reasons for the model's performance differences on AgentClinic-MedQA and MedQA.
- Although the article introduces human scoring of the model's performance, it does not involve humans in the simulation experiments, making it difficult to measure the differences between the model's performance and human performance.

**Questions:**

- Why is a measurement agent needed, and how does its role differ from that of a tool? I hope the author can provide a deeper explanation.
- It seems that the doctor agent in AgentClinic-MedQA has the opportunity to gather more information through multiple rounds of interaction compared to the doctor agent in MedQA. However, in actual tests, the performance of the doctor agent in AgentClinic-MedQA may be inferior to that of the doctor agent in MedQA. There could be many reasons for this phenomenon. Can the author provide an intermediate metric, such as the amount of useful information obtained by the doctor agents in both AgentClinic-MedQA and MedQA, to further determine the causes of this phenomenon?
- Is it possible to have human doctors interact with simulated patients under the most basic settings and then compare the performance differences between human doctors and LLM doctors? This might provide a more intuitive comparison.

---

> ### Author Response · Authors · 2024-11-24
>
> We thank the reviewer for their insightful feedback and for suggesting analyses that have improved our work. Below, we provide a detailed, point-by-point response to each comment:
>
> > 1 & 2. Can the author provide an intermediate metric, such as the amount of useful information obtained by the doctor agents in both AgentClinic-MedQA and MedQA, to further determine the causes of this phenomenon? The article does not conduct an in-depth analysis of the reasons for the model's performance differences on AgentClinic-MedQA and MedQA.
>
> We thank you for the suggestion and believe this analysis has improved the quality of our work. At your suggestion, we ran a sample of MedQA and AgentClinic-MedQA cases side-by-side to find out the average information coverage by the doctor agent. To do this, we collect all of the relevant information from the MedQA case (e.g. the patient has difficulty walking, the patient has back pain, etc). We then ran the AgentClinic-MedQA simulation with GPT-4 on these cases and manually reviewed the dialogue in order to mark each information item as covered if the doctor agent was able to extract this information from the patient agent from dialogue or through the measurement agent. We find that the average amount of information in the original MedQA question that is revealed (denoted coverage) is 67% We find that in cases where the doctor agent misdiagnoses the patient, the average coverage is 63% and in cases where the doctor agent provides a correct diagnosis, the coverage is 72%. We believe that it is for this reason that there is a discrepancy between MedQA and AgentClinic-MedQA performance–in MedQA all of the information is provided up front, whereas in AgentClinic-MedQA the LLM must extract the information by interacting with the environment under uncertainty, ultimately making a decision with less information. The details of this experiment are in Appendix F.3.
>
> > 3. The article's simulation of interactions between multiple agents is relatively simple, with the main interactions in the experiment limited to between the doctor agent and the patient agent.
>
> We thank you for this observation. We would like to clarify that the primary goal of AgentClinic is not to create simulated environments identical to real-world settings. Instead, our objective is to demonstrate the impact of transitioning from static medical QA benchmarks to a dynamic, interactive framework. By "bringing the agents to life," where the medical LLM must actively collect information, order lab tests, and perform sequential decision-making, we highlight the significant discrepancies in performance compared to static evaluations. This shift provides a more nuanced assessment of an LLM's capabilities, demonstrating the complexities of sequential clinical reasoning over static question-answering scenarios.
>
> > 4. Why is a measurement agent needed, and how does its role differ from that of a tool? I hope the author can provide a deeper explanation.
>
> The reason why there is a measurement agent is that the doctor agent is not allowed to have access to all retrospectively collected data to properly simulate that the doctor requests a given imaging or lab test. The difference between an agent and a tool would be that the tool would need a highly specific instruction, whereas by creating a measurement agent, we can have a flexible interface where different LLMs which may have different styles of requesting measurements (different acronyms and word ordering) and it will be interpreted with less difficulty. However, whether it is a measurement or database (DB) agent or simply a tool to be called, is in the end arbitrary - both can be used to allow the doctor agent to collect measurement results. It is possible that we can add the ablation, seeing what happens if all the data handling is operated as a tool - although we do not believe it will be of high relevance to the main takeaways of the paper. We discussed with clinicians who found the measurement / DB agent relevant in that there are agent systems in development (say copilots) that can be chatted to to retrieve specific patient information in the EHR.

---

> > ### Author Response · Authors · 2024-11-24
> >
> > > 5. Is it possible to have human doctors interact with simulated patients under the most basic settings and then compare the performance differences between human doctors and LLM doctors? & 6. Although the article introduces human scoring of the model's performance, it does not involve humans in the simulation experiments, making it difficult to measure the differences between the model's performance and human performance.
> >
> > We thank you for the suggestion and believe this is a great idea. To address this, we conducted an experiment where three physicians acted as the doctor agent within the AgentClinic framework. These physicians interacted with the patient agent and measurement agent under the same conditions as the LLM-based doctor agent, providing diagnoses based on the simulated cases. Their performance was evaluated using the same scoring metrics applied to the LLMs. The observed accuracy on AgentClinic-MedQA for the physicians was 54 ± 28.5%, offering a direct comparison to the performance of LLM-based doctor agents. We note that this was incredibly time consuming for our physicians to gather, and thus were only able to evaluate on 50 cases from 3 physicians. We will be able to extend this to more cases in our revisions. Nonetheless, we believe this addition provides valuable insights into the performance of humans versus LLMs in AgentClinic when compared to expert humans.

---

> ### Comment · Reviewer_6dqE · 2024-11-26
>
> The authors' response effectively addressed my concerns, and I have accordingly raised the score.

---

### Official Review · Reviewer_iemz · 2024-11-04

**Soundness:** 3
**Presentation:** 3
**Contribution:** 3
**Rating:** 8
**Confidence:** 4

**Summary:**

The paper introduces AgentClinic, a multimodal benchmark that evaluates language models' medical diagnostic abilities through **interactive dialogue** and examinations rather than static questions. Using four agents and incorporating biases, tools, and real clinical cases across specialties and languages, it demonstrates Claude-3.5's superior performance while revealing varied capabilities among models in *tool usage* and *bias handling*.

**Strengths:**

Overall, the paper is well-written with clear logical flow and is easy to follow. The motivation is clearly presented and makes sense: "Existing diagnostic challenges are not static QAs, but are interactive, dialogue-driven, sequential decision-making environments that require data collection, ordering appropriate medical exams, and understanding medical images across patients with unique family histories, lifestyle habits, age categories, and diseases." This naturally addresses key limitations in previous medical LLM agents' static QA-oriented tasks. The human evaluations from experts make the presented experiments solid. The discussion is also impressive and comprehensive.

**Weaknesses:**

Weaknesses discussed in the Discussion section:
- The simulated clinical environments are currently simple; more methods could be benchmarked:
  1. Encouraging Divergent Thinking in Large Language Models through Multi-Agent Debate (https://arxiv.org/abs/2305.19118)
  2. MedAgents (https://aclanthology.org/2024.findings-acl.33/)
  3. ReConcile (https://arxiv.org/abs/2309.13007)
- Uncertainty issues

Suggestions for improvement:
- Include a statistics table for utilized/built datasets, covering sample size, dataset modalities included, task types/descriptions
- Address ethics issues with closed-source LLMs by replacing them with open-source LLMs running offline to prevent patient-sensitive data leakage
- Fix typography: use ``xxx'' in LaTeX for quotes
- Add more multi-agent collaboration/MDT baselines
- Evaluate sensitivity to various prompting strategies

**Questions:**

See Weaknesses

**Details Of Ethics Concerns:**

Note: Closed-source models such as GPT-4 are prohibited for MIMIC-III and MIMIC-IV according to their data use policy (https://physionet.org/about/licenses/physionet-credentialed-health-data-license-150)

---

> ### Author Response · Authors · 2024-11-22
>
> We thank the reviewer for insightful feedback and for noting the paper is well-written, the experiments are solid, and that the discussion is impressive and comprehensive. The reviewer asks us great questions which we provide detailed answers below.
>
> > *1. “Uncertainty issues”*
>
> We appreciate your concern regarding uncertainty, particularly the possibility that models might be overfitted to datasets like MedQA, which could potentially diminish the significance of high AgentClinic-MedQA accuracy. This is a valid and thoughtful observation. However, we argue that what might be perceived as a potential limitation is actually one of our most significant findings. While a model's training on MedQA could raise questions about the significance of its high accuracy, our results show that the performance gap between AgentClinic and MedQA is independent of any potential overfitting. To substantiate this, we present results from multiple models—such as Mixtral-8x7B, Llama 2 70B, and Llama 3 70B—which are widely recognized as not being overfit on MedQA.
>
> These findings suggest that regardless of a model's prior knowledge of MedQA answers, our main conclusions remain robust and unaffected by overfitting concerns. In other words, even if a model "knows" the answer to a MedQA question, the dynamic nature of AgentClinic—requiring sequential decision-making, such as interacting with the patient to gather information and engaging with measurement agents to obtain medical results—makes mere knowledge of the answer far less valuable than the model's ability to reason and act effectively within the environment.
>
> > *2. “Include a statistics table for utilized/built datasets, covering sample size, dataset modalities included, task types/descriptions”*
>
> Thank you for this valuable suggestion. We have incorporated the requested statistics and added detailed tables to the manuscript. Table 1 provides a breakdown of dataset modalities along with accuracy metrics by modality, while Table 2 offers comprehensive dataset descriptions. To further improve transparency, we have included the exact generation files we used at the following de-identified link: https://github.com/AgentClinic/AgentClinicAnonymized/.
>
> Table #1
>
> | **Category** | **n** | **% of imgs** | **GPT-4 %** | **GPT-4o %** | **GPT-4o-mini %** |
> |--------------------|------------------|------------------|-------------|--------------|-------------------|
> | Physical | 56 | 42% | 31.4 | 15.7 | 11.1 | | CT | 19, 16% | 26.3 | 10.5 | 0 |
> | Dermatology | 16 | 13% | 37.5 | 6.3 | 7.6 |
> | Hist/Path | 13 | 11% | 15.3 | 15.3 | 9 |
> | Radiography | 12 | 10% | 0 | 8.3 | 0 |
> | Ophthalmology | 11 | 9% | 27.2 | 27.2 | 0 |
> | MRI | 6 | 5% | 0 | 16.7 | 0 |
> | Biopsy | 6 | 5% | 50 | 33.3 | 33.3 |
> | Surgery | 3 | 3% | 33.3 | 0 | 50 |
> | Instrument | 2 | 2% | 50 | 50 | 0 |
> | ECG | 2 | 2% | 50 | 0 | 0 |
> | Echocardiogram | 2 | 1% | 100 | 0 | 0 |
> | Ultrasound | 1 | 1% | 0 | 0 | 0 |
>
> Table #2
> | **Dataset Name**        | **Sample Size**                             | **Modalities Included**   | **Task Types/Descriptions**                                                                                   |
> |--------------------------|---------------------------------------------|----------------------------|---------------------------------------------------------------------------------------------------------------|
> | AgentClinic-NEJM         | 120 cases derived from NEJM case challenges | Multimodal (Text + Images) | Open-ended diagnostic tasks requiring image analysis and patient dialogues.                                   |
> | AgentClinic-MedQA        | 215 cases derived from USMLE case challenges| Text                      | Simulated cases with structured patient information from USMLE data.                                         |
> | AgentClinic-MIMIC-IV     | 200 cases derived from MIMIC-IV             | Text                      | Simulated cases with structured patient information from real-world EHR data.                                |
> | AgentClinic-Spec         | 260 cases derived from from MedMCQA                          | Text                      | Specialist diagnostic cases from 9 medical specialties, including pediatrics, psychiatry, and internal medicine. |
> | AgentClinic-Lang         | 749 cases derived from AgentClinic-MedQA              | Multilingual Text         | AgentClinic-MedQA cases translated for 7 languages (English, Chinese, Hindi, Korean, Spanish, French, Persian). |

---

> > ### Author Response · Authors · 2024-11-22
> >
> > > *3. Address ethics issues with closed-source LLMs by replacing them with open-source LLMs running offline to prevent patient-sensitive data leakage.*
> >
> > We conducted our experiments with MIMIC-IV data in strict compliance with the PhysioNet guidelines on the “Responsible Use of MIMIC Data with Online Services like GPT” (PhysioNet Guidelines). This approach permits the use of closed-source models through secure platforms, such as Azure for OpenAI services and Amazon Bedrock for Claude-3.5-Sonnet. Additionally, all open-source models were run locally to ensure full control over the inference process.
> >
> > > *4. Fix typography: use ``xxx'' in LaTeX for quotes*
> >
> > Thank you very much, we have updated the quotes in our manuscript.
> >
> > > *5. The simulated clinical environments are currently simple; more methods could be benchmarked” and 6. Add more multi-agent collaboration/MDT baselines*
> >
> > We appreciate your suggestions and have taken steps to address them. To expand the scope of our benchmarks, we have included results for new methods. Additionally, we agree that multi-agent collaboration is an intriguing avenue to explore. In this regard, we have collected results for both Multi-Agent Debate and MedAgents. Below are the results:
> > * Multi-Agent Debate (gpt-4)  51.7% ± 3.0
> > * Multi-Agent Debate (gpt-4o)  37.9% ± 3.1
> > * Multi-Agent Debate (claude-3.5-sonnet)  64.1% ± 3.4
> > * MedAgents (gpt-4)  53.1% ± 3.1
> > * MedAgents (gpt-4o)  40.1% ± 3.3
> > * MedAgents (claude-3.5-sonnet)  65.2% ± 3.6
> >
> > > The simulated clinical environments are currently simple
> >
> > We want to clarify that the primary goal of AgentClinic is not to create simulated environments that are identical to real-world clinical scenarios. Instead, our focus is on demonstrating that transitioning from static medical QA benchmarks to dynamic environments—where agents must gather information, order lab tests, and make sequential decisions—reveals significant performance discrepancies.
> >
> > This is evident in our findings, where the current state-of-the-art LLMs achieve considerably lower performance on our AgentClinic-MedQA benchmark (62.1%) compared to their performance on the static MedQA benchmark (91.1%) for the same problems. Additionally, much work is particularly needed on the multimodal environments, where the best SOTA models are shown to obtain a highest score of 37.2%.
> >
> > > Evaluate sensitivity to various prompting strategies
> >
> > Thank you for this insightful suggestion. We recognize the importance of this aspect in understanding model performance. Inherently, the AgentClinic framework involves diverse prompting scenarios through its multimodal and interactive environment. For example, interactions with tools and agents, such as the measurement agent or the patient agent, require varied and dynamic prompts to guide the model in retrieving and interpreting information effectively. These interactions test the models’ ability to handle different prompt formulations within the context of sequential decision-making and multimodal data. While this was not a primary focus of our study, the performance differences observed across models may partly reflect their sensitivity to the complex prompts required by the benchmark.

---

> > > ### Comment · Reviewer_iemz · 2024-11-25
> > >
> > > Thank you for your response - my concerns have been resolved. I appreciate your work and have updated my score to a clear accept.

---

### Official Review · Reviewer_mXnj · 2024-11-10

**Soundness:** 3
**Presentation:** 3
**Contribution:** 3
**Rating:** 8
**Confidence:** 3

**Summary:**

Recognizing that existing static QA benchmarks often fail to reflect the complexities of clinical decision-making tasks, the authors propose AgentClinic, an open-source multimodal agent benchmark for simulating clinical environments. The framework introduces patient agents (informed by real clinical cases), doctor agents, a measurement agent, and a moderator, with agents exhibiting 24 different biases. The patient cases represent nine medical specialities across seven multilingual environments. The study reveals significant performance differences across models, highlighting the impact of biases and tool integration, with models like Claude-3.5 and Llama-3 demonstrating notable improvements with tools like adaptive retrieval and note-taking.

**Strengths:**

Dataset: This is a novel and comprehensive benchmark that closely mimics physician-patient interactions. Its inclusion of diverse medical scenarios ensures a comprehensive assessment of model capabilities, providing a challenging benchmark for evaluating model accuracy and generalization.

Approach: The authors use a robust benchmarking approach that includes multiple state-of-the-art models (e.g., Claude 3.5, GPT-4, Mixtral-8x7B, Llama 3, etc.) evaluated on the same task, and quantify uncertainty in performance. This methodology ensures a fair and statistically rigorous comparison of the models, enhancing the reliability and transparency of the results.

Experiment: The human evaluation ratings provided by physicians adds a layer of real-world applicability of this benchmark.

**Weaknesses:**

Stigmatizing language in medical records can influence not only how physicians perceive patients, but also how treatment decisions are made. Physicians have been found to prescribe pain medication less often when patient notes contain stigmatizing versus neutral language (P Goddu et al., 2018; Kelly et al., 2010). This dynamic can influence treatment outcomes and the overall patient-provider relationship. The clinical cases in the paper could benefit from more nuanced language that reflects these concerns, particularly in how bias is conveyed through medical documentation. Additionally, the paper’s "Bias and Patient Agent Perception" section could be expanded to explore patient trust in the healthcare system.

**Questions:**

1. What is the rationale behind an N of 20 for the interaction time when assessing the diagnostic accuracy of AgentClinic-MedQA? I’m surprised at the meaningful drop in diagnostic accuracy from 52% to 25%.

2. Although Claude 3.5 is reported to have the highest accuracy on both AgentClinic-MIMIC-IV and AgentClinic-MedQA, its performance varies by as much as 13 percentage points. What factors contribute to this moderate variability? Does this suggest issues with model robustness or potential limitations in the dataset or experimental design?

3. Have the authors conducted a detailed error analysis to identify the types of questions/ cases the models struggle with the most? Are there specific failure modes common to all the models, or do different models exhibit distinct weaknesses?

---

> ### Author Response · Authors · 2024-11-22
>
> We thank the reviewer for acknowledging the novelty of our work and for noting its relevance to medical AI and its real-world applicability. We have made improvements based on your insightful feedback which is addressed point-by-point below.
>
> > *1. What is the rationale behind an N of 20 for the interaction time when assessing the diagnostic accuracy of AgentClinic-MedQA? I’m surprised at the meaningful drop in diagnostic accuracy from 52% to 25%.*
>
> One of the variables that can be changed during AgentClinic evaluations is the amount of interaction steps (N) that the doctor is allotted. In initial experiments, we discovered that the doctor agent’s diagnostic performance is surprisingly sensitive to the number of interaction steps the agent has access to. When allowed, the doctor agent will continue asking questions without any foreseeable stopping point. We thus determined it was necessary to find a reasonable value for N, which is the way we arrived at our rationale for choosing N=20 for the interaction time in our experiments. In order to determine a value to select for N, we studied varying the number of interaction steps through our experiments shown in Appendix F.1 “How does limited time affect diagnostic accuracy.” We note from Section L.1 “Doctor Agent Instructions” that the doctor is reminded each step how many interactions they have left. When varied (e.g. N=10, N=20, or N=30) the doctor agent adjusts to the new value and is not optimized for a particular N in any way, but rather can flexibly adapt to any value N that is given.
>
> **N < 20**: We find that on AgentClinic-MedQA accuracy with GPT-4 decreases from 52% when N=20 to 25% when N=10 and 38% when N=15, showing a near-linear relationship between interactions allotted and diagnostic accuracy (~13 accuracy points per 5 interactions).
>
> **N > 20**: When N is set to a larger value, N=25 and N=30, the accuracy actually decreases slightly from 52% when N=20 to 48% when N=25 and 43% when N=30. The reason that increasing N to N=30 decreases accuracy is likely due to LLMs' tendency to focus on the first and last parts of large contexts, neglecting the middle [1]. Future work on AgentClinic could incorporate a short-term memory module, as seen in previous agent work, to better manage context and potentially improve accuracy as N increases.
>
> [1] Liu, Nelson F., et al. "Lost in the middle: How language models use long contexts." Transactions of the Association for Computational Linguistics (2024).
>
> > *2. Although Claude 3.5 is reported to have the highest accuracy on both AgentClinic-MIMIC-IV and AgentClinic-MedQA, its performance varies by as much as 13 percentage points.*
>
> The performance of Claude-3.5-Sonnet on AgentClinic-MedQA is 62.1% and on AgentClinic-MIMIC-IV is 42.9% showing a 19.2% discrepancy. We also note that the average score across all LLMs on AgentClinic-MedQA is 35.1% and AgentClinic-MIMIC-IV is 25.7% (9.4 points lower). We also note the average relative reduction in performance [i.e. (acc_1 - acc_2)/acc_1] from MedQA to MIMIC-IV is 32.6%, from which Claude-3.5-Sonnet is 30.9% different, which means that it is actually showing a relative performance decrease aligned with the other models.
>
> The reason that there is an accuracy difference between MedQA and MIMIC-IV cases in general is due to the inherent difficulty of MIMIC-IV cases versus the MedQA cases.  The factors contributing to this is that in MedQA, cases are based on the USMLE, which are synthetically designed by medical board examiners for medical students to be simpler and more common cases, whereas MIMIC-IV is from real medical cases using real data which is a lot more challenging. This is also reflected in existing MIMIC-IV-based QA benchmarks, where USMLE-based/MedQA questions are shown to be easier for LLMs to solve than MIMIC-IV questions [1, 2].
>
> [1] Bae, Seongsu, et al. "Ehrxqa: A multi-modal question answering dataset for electronic health records with chest x-ray images." Advances in Neural Information Processing Systems 36 (2024).
>
> [2] Nori, Harsha, et al. "Can generalist foundation models outcompete special-purpose tuning? case study in medicine." arXiv preprint arXiv:2311.16452 (2023).
>
> > *3. The paper’s "Bias and Patient Agent Perception" section could be expanded to explore patient trust in the healthcare system*
>
> We appreciate the suggestion to expand the "Bias and Patient Agent Perception" section by exploring patient trust in the healthcare system. While this is an intriguing area of study, the primary focus of our work is on advancing the doctor agent. This is because the ultimate goal is to deploy the doctor agent in hospital settings, where it can directly contribute to advancing medical practice and improving patient outcomes.That said, if preferred, we could include follow-up experiments with the patient agent to examine their trust in the healthcare system, in line with similar experiments presented in this section.

---

> > ### Author Response · Authors · 2024-11-22
> >
> > > *4. Have the authors conducted a detailed error analysis to identify the types of questions/ cases the models struggle with the most? Are there specific failure modes common to all the models, or do different models exhibit distinct weaknesses?*
> >
> > Yes. In order to figure this out, we ran a sample of MedQA and AgentClinic-MedQA cases side-by-side to find out the average information coverage by the doctor agent. To do this, we collect all of the relevant information from the MedQA case (e.g. the patient has difficulty walking, the patient has back pain, etc). We then ran the AgentClinic-MedQA simulation with GPT-4 on these cases and manually reviewed the dialogue in order to mark each information item as covered if the doctor agent was able to extract this information from the patient agent from dialogue or through the measurement agent. We find that the average amount of information in the original MedQA question that is revealed (denoted coverage) is 67% We find that in cases where the doctor agent misdiagnoses the patient, the average coverage is 63% and in cases where the doctor agent provides a correct diagnosis, the coverage is 72%. We found that lower coverage was associated with cases where there were many symptoms that were unrelated to the actual diagnosis, leading the doctor down the wrong decision pathway. Additionally, for multimodal questions, we find that all models consistently struggle to interpret images from the following modalities: Ultrasound (0%), Radiograph (2.7%), MRI (5.6%), Hist/Path (10.2%) where % here represent accuracy averages. These numbers are reported in Table 1.

---

> > > ### Comment · Reviewer_mXnj · 2024-11-25
> > >
> > > Thank you for your response!

---

### Official Review · Reviewer_LHBV · 2024-11-29

**Soundness:** 3
**Presentation:** 4
**Contribution:** 3
**Rating:** 6
**Confidence:** 5

**Summary:**

This paper presents AgentClinic, a multimodal agent benchmark for evaluating large language models (LLMs) in simulated clinical environments. It challenges the traditional static question-answering evaluations by introducing interactive, dialogue-driven, sequential decision-making scenarios. The benchmark includes patient interactions, multimodal data collection, and tool usage, covering nine medical specialties and seven languages.

**Strengths:**

The paper presents an approach to evaluating LLMs in clinical environments by introducing AgentClinic, a multimodal agent benchmark. This is a departure from the traditional static question-answering evaluations and provides a more realistic and comprehensive assessment of LLMs' capabilities in medical diagnosis. The incorporation of biases into the benchmark is an original contribution, as it allows for the study of how biases can affect the performance of LLMs and patient perception.

The paper is well-written and easy to follow, with clear explanations of the benchmark design, agent roles, and evaluation metrics. The figures and tables are also well-designed and help to illustrate the key findings.

The work may has implications for the development and evaluation of medical AI systems. The study of biases in clinical environments is also of great importance, as it can help to improve the fairness and reliability of medical AI systems.

**Weaknesses:**

The study does not consider the impact of longitudinal data on the performance of LLMs, which is an important aspect of clinical decision-making.

While the paper introduces several novel evaluation metrics, such as patient compliance and consultation ratings, these metrics may be subjective and difficult to measure accurately.

One of my main concerns is that the core framework of this article seems to have a large overlap with AI Hospital (which also uses a multi-agent framework to evaluate the ability of doctor agents in multi-round medical interactions), but this article does not mention AI Hospital at all.

Ref: AI Hospital: Benchmarking Large Language Models in a Multi-agent Medical Interaction Simulator

**Questions:**

1. How do you plan to address the issue of potential bias in the training data of proprietary models like GPT-4 and Claude 3.5? Can you provide more details on the steps you have taken to mitigate this bias?
2. The simulated clinical environment in AgentClinic seems to be relatively simple compared to real-world clinical settings. How do you plan to expand and improve the benchmark to better capture the complexity of actual clinical practice?
3. The evaluation metrics used in the paper, such as patient compliance and consultation ratings, are subjective. How do you plan to validate and improve the reliability of these metrics?
4. From the perspective of AI Hospital, some of the core contributions claimed by this paper will be greatly weakened, and I wonder if the authors can explain why AI Hospital is not cited. Alternatively, can you explain the similarities and differences between this article's framework and the AI ​​Hospital framework?

---

> ### Author Response · Authors · 2024-11-30
>
> We thank the reviewer for acknowledging the paper is well-written, our findings on biases are of great importance, and that the key findings are well illustrated. We note that this review was submitted after the review period (by 26 days) and after the revision period (3 days), but will do our best to answer the points raised over the holiday weekend.
>
> > 1. The study does not consider the impact of longitudinal data on the performance of LLMs, which is an important aspect of clinical decision-making.
>
> We appreciate the suggestion to include longitudinal data. However, the primary focus of this study is on transitioning from static, complete information case scenarios to dynamic, incomplete information scenarios that better simulate real-world clinical interactions. This focus aims to address a different gap in the literature.
>
> > 2 and 3. While the paper introduces several novel evaluation metrics, such as patient compliance and consultation ratings, these metrics may be subjective and difficult to measure accurately. The evaluation metrics used in the paper, such as patient compliance and consultation ratings, are subjective. How do you plan to validate and improve the reliability of these metrics?
>
> We appreciate the reviewer’s concern regarding the subjectivity of metrics like patient compliance and consultation ratings. While compliance is self-assessed by the patient agent, it has the potential to be objectified in future work by analyzing correlations between self-reported compliance and actual adherence to diagnoses and treatment plans in real-world settings. Subjective metrics, such as consultation ratings, are equally important because they capture patient experience, a critical yet underrepresented aspect of clinical care. Historically, clinical evaluations have prioritized quantifiable outcomes like survival rates, often overlooking the patient’s perspective. By integrating these metrics, we aim to promote empathetic, patient-centered AI systems.
>
> > 4 and 5. One of my main concerns is that the core framework of this article seems to have a large overlap with AI Hospital, but this article does not mention AI Hospital at all. From the perspective of AI Hospital, some of the core contributions claimed by this paper will be greatly weakened, and I wonder if the authors can explain why AI Hospital is not cited. Alternatively, can you explain the similarities and differences between this article's framework and the AI ​​Hospital framework?
>
> We thank the reviewer for their questions regarding AI Hospital and appreciate the opportunity to clarify the distinctions between the two studies. While both works explore multi-agent systems in medical contexts, they address different research questions and have distinct contributions:
>
> 1. AgentClinic is specifically designed to investigate how LLMs transition from static QA benchmarks like MedQA to dynamic, sequential decision-making scenarios under incomplete information. By requiring models to engage with patients, gather relevant details, and request appropriate tests, AgentClinic reveals novel insights, such as the significant drop in performance when static cases are tackled interactively. AI Hospital does not focus on this problem or enable such comparisons.
>
> 2. AgentClinic introduces multimodal capabilities, integrating text, images, and lab results, to better reflect real-world clinical workflows. In contrast, AI Hospital primarily focuses on textual data from Chinese medical records, offering a narrower scope of evaluation.
>
> 3. Our framework systematically incorporates and studies implicit and explicit biases in individual agents, such as patient biases or diagnostic tendencies, to evaluate their effects on outcomes. This aspect is not explored in AI Hospital.
>
> 4. AgentClinic equips doctor agents with advanced tools, including adaptive retrieval systems, persistent note-taking, and reasoning chains, to emulate realistic clinical reasoning processes. AI Hospital does not feature comparable tool support or techniques for enhancing agent performance.
>
> 5. AgentClinic evaluates multilingual capabilities across seven languages, including English, Chinese, Spanish, and Persian, enabling global applicability. AI Hospital is focused exclusively on Chinese medical records and does not assess multilingual performance.
>
> 6. AgentClinic includes detailed human evaluations by clinical experts, encompassing both offline assessments of agent interactions and live role-playing scenarios where experts act as the doctor. In contrast, AI Hospital does not report quantifiable results from human evaluations. The supplementary materials of AI Hospital include a review form used by medical students (likely Supplementary Figure 8), but no actual results or performance metrics from these reviews are provided. This lack of detailed human validation limits the insights that can be drawn about the real-world applicability of AI Hospital’s framework compared to AgentClinic.

---

> > ### Author Response · Authors · 2024-11-30
> >
> > > 6. Ref: AI Hospital: Benchmarking Large Language Models in a Multi-agent Medical Interaction Simulator
> >
> > We would be glad to cite and better highlight the differences between AI Hospital and AgentClinic in our related works, but would like to note that this review was posted after the revision period was over so we are unable to provide revisions to the current manuscript. We would be glad to add this to the related works for the camera-ready paper. We also note that we highlight many other related works in our manuscript [1-5] which are also of high relevance.
> >
> > [1] Tao Tu, Anil Palepu, Mike Schaekermann, Khaled Saab, Jan Freyberg, Ryutaro Tanno, Amy Wang, Brenna Li, Mohamed Amin, Nenad Tomasev, et al. Towards conversational diagnostic ai. arXiv preprint arXiv:2401.05654, 2024.
> >
> > [2] Shreya Johri, Jaehwan Jeong, Benjamin A Tran, Daniel I Schlessinger, Shannon Wongvibulsin, Zhuo Ran Cai, Roxana Daneshjou, and Pranav Rajpurkar. Guidelines for rigorous evaluation of clinical llms for conversational reasoning. medRxiv, pp. 2023–09, 2023.
> >
> > [3] Xiangru Tang, Anni Zou, Zhuosheng Zhang, Yilun Zhao, Xingyao Zhang, Arman Cohan, and Mark Gerstein. Medagents: Large language models as collaborators for zero-shot medical reasoning. arXiv preprint arXiv:2311.10537, 2023.
> >
> > [4] Stephen R Pfohl, Heather Cole-Lewis, Rory Sayres, Darlene Neal, Mercy Asiedu, Awa Dieng, Nenad Tomasev, Qazi Mamunur Rashid, Shekoofeh Azizi, Negar Rostamzadeh, et al. A toolbox for surfacing health equity harms and biases in large language models. arXiv preprint arXiv:2403.12025, 2024.
> >
> > [5] Yusheng Liao, Yutong Meng, Yuhao Wang, Hongcheng Liu, Yanfeng Wang, and Yu Wang. Automatic interactive evaluation for large language models with state aware patient simulator. arXiv preprint arXiv:2403.08495, 2024.
> >
> > > 7. How do you plan to address the issue of potential bias in the training data of proprietary models like GPT-4 and Claude 3.5? Can you provide more details on the steps you have taken to mitigate this bias?
> >
> > It is possible that models like GPT-4 and Claude 3.5 were trained on datasets such as MedQA, potentially ("biasing"/) benefiting from data leakage--this is addressed in our work. One of our main findings demonstrate that performance on the datasets included in the model's train set (MedQA) does not reliably predict accuracy on the dynamic AgentClinic form (AgentClinic-MedQA) of those questions (see Figure 3), suggesting that reliance on prior training data alone is insufficient for success in dynamic, interactive scenarios. Notably, models like Mixtral-8x7B and Llama 2-70BChat have not reported training on MedQA’s test or train sets, providing an important comparative baseline.
> >
> > > 8. The simulated clinical environment in AgentClinic seems to be relatively simple compared to real-world clinical settings. How do you plan to expand and improve the benchmark to better capture the complexity of actual clinical practice?
> >
> > The primary aim of AgentClinic is not to fully replicate real-world clinical environments but to highlight the challenges LLMs face when transitioning from static QA benchmarks to dynamic, interactive scenarios requiring agents to gather information, order tests, and make sequential decisions under uncertainty. Our findings reveal that state-of-the-art LLMs perform significantly worse on AgentClinic-MedQA (62.1%) compared to static MedQA (91.1%) for identical cases, demonstrating the impact of interactive settings. Additionally, in multimodal scenarios, top models achieve only 37.2% which demonstrates the need for progress in handling complex clinical tasks.
> >
> > Regarding future work, we outline this in the discussion section. Future work aims to expand the simulated clinical environments by incorporating additional critical roles, such as nurses, patients' relatives, administrators, and insurance contacts, to better reflect the complexities of real-world healthcare systems. Plans also include integrating agents into simulated physical environments to account for factors like hospital space constraints and examining the influence of demographic biases, including race and gender, on clinical decision-making.

---

> > ### Comment · Reviewer_LHBV · 2024-12-02
> >
> > I appreciate the author's patient response, but I carefully checked AI Hospital and some other related works, the author does not seem to have fully understood it correctly. In fact, AI Hospital is also dealing with dynamic, sequential decision-making scenarios with incomplete information. However, AI Hospital is based on the real-world medical records of patients, and doctors also need to collect more symptoms of patients and recommend appropriate medical examinations in multi-turn interaction.
> >
> > As for the conclusion, it is actually similar to AgentClinic. AI Hospital also reveals that the effect of interactive scenarios is far inferior to that of non-interactive scenarios (i.e. one-step). Therefore, I think that this article's transition from static QA benchmarks such as MedQA to dynamic ones does not seem to be a novel approach.
> >
> > In fact, traditional work should also be cited (there are many), These works are all classic studies in the field of automatic diagnosis, and they also deal with dynamic medical decision-making.  For example:
> > 1. End-to-end knowledge-routed relational dialogue system for automatic diagnosis
> > 2. Task-oriented Dialogue System for Automatic Disease Diagnosis via Hierarchical Reinforcement Learning
> >
> > Overall, my main concern is that AgentClinic should not ignore these related studies in terms of motivation and innovation. I acknowledge the contributions of AgentClinic and believe that AgentClinic deserves to appear in ICLR. However, the authors should not skip over these foundational works and claim the transition from QA benchmarks to dynamics as their own unique contribution.
> >
> > As for the bias, multimodality, tool calling and multilingualism mentioned by the author, they are not uncommon in general fields. I hope that the author can be more inclusive in discussing relevant foundational technologies, at least in the medical field, in subsequent versions.
> >
> > I am willing to continue the discussion with the author, but will reserve the intention of raising the score for now.

---

> > > ### Author Response · Authors · 2024-12-02
> > >
> > > We appreciate the reviewer's analysis and the opportunity to clarify distinctions between AgentClinic and prior works, including AI Hospital. While there are thematic similarities, fundamental differences exist in methodology, scope, and innovation that strongly validates the originality of AgentClinic. We again note the significant differences between these two works:
> > >
> > >
> > > 1. **Clinical validation**: AgentClinic includes **comprehensive evaluations by three clinical experts (MDs), both offline and online** through live role-playing studies, providing richer and more granular insights into model behavior and clinical utility. In contrast, AI Hospital reports a student review form in its supplementary material but does not include quantifiable results, strongly limiting its evaluation depth, requiring the reader to believe that the students validated the chats without any quality scores provided.
> > >
> > > 2. **Comparison to Static Benchmarks**: AI Hospital includes interactive scenarios, but its one-step evaluation approach is not clearly described (e.g., specifics on prompts and context). Additionally, the challenge posed by its MVME benchmark is unclear, as models like GPT-4 reportedly achieve near-perfect scores on several tasks. AI Hospital does not provide comparisons to established benchmarks. In contrast, **AgentClinic highlights the limitations of widely-used benchmarks like MedQA, showcasing performance collapse**—up to 10x drop in diagnostic accuracy—when transitioning to dynamic equivalents. This underscores the value of direct comparisons with established benchmarks, revealing critical gaps in the literature that bespoke datasets currently overlook.
> > >
> > > 3. **Scope of Evaluation**: While AI Hospital addresses multi-turn interactions, it focuses on textual data from Chinese medical records. In contrast, AgentClinic evaluates multimodal capabilities (e.g., text, real medical images, biases, multilingual text, agent tools, and lab results) to better reflect the integration of diverse data streams central to real-world clinical workflows
> > >
> > > 4. **Tool Usage and Advanced Agent Capabilities**: AgentClinic introduces tools such as **persistent note-taking, adaptive retrieval systems, and reflective reasoning cycles**, designed to mimic and evaluate key aspects of clinical reasoning. These tools are rigorously studied to assess their impact on performance across multiple tasks, representing a novel contribution not present in AI Hospital.
> > >
> > > 5. **Bias and Diversity**: AgentClinic explicitly models biases and evaluates patient-centric metrics while supporting multilingual evaluation across seven languages (e.g., English, Chinese, Spanish). This focus on global applicability and fairness significantly broadens its scope compared to AI Hospital, which evaluates performance on Chinese datasets.
> > >
> > > We appreciate the reviewer's suggestions and will gladly cite AI Hospital and related foundational works in the camera-ready version to better contextualize our contributions. However, we respectfully emphasize that AgentClinic's focus on clinical validation, multimodal interactions, advanced tool use, multilingual evaluations, and explicit benchmarking against widely-used and field-established datasets represents a distinct and valuable addition to the field.

---

> > > > ### Comment · Reviewer_LHBV · 2024-12-03
> > > >
> > > > The author has addressed some of my concerns, but based on some of the author's above claims, I don't think the authors really understand AI Hospital. I have also reviewed this work, so I see a lot of familiar similarities in the framework.
> > > >
> > > > The paradigm of multi-agents has always existed in the medical field. For example, agent-based approaches to modeling the spread of infectious diseases have existed since the 1990s. In early studies, many agents were rule-based, difficult to interact with, and lacked much flexibility due to insufficient model capabilities. Using LLMs to simulate various possible roles in the medical environment is a natural transition. Agent Hospital, AgentClinic, AI Hospital, these works are all similar attempts, with not exactly the same purpose.
> > > >
> > > > What I would like to see is that the author can develop your story along the lines of Agent in Medical in the revised version, so that people can make your contribution clear. The main motivation for my previous comments is that the claimed contributions and innovations of AgentClinic are exaggerated (At least that's what I think). However, given the completeness and comprehensiveness of the work, I have updated the score.

---

### Meta-Review · Area_Chair_PWjf · 2024-12-23

**Metareview:**

*Note: This decision was confirmed by both the Senior Area Chair and the Area Chair.*

The paper introduces AgentClinic, a benchmark that uses multiple agents to make medical decisions. Specifically, the paper creates four LLMs which interact as a doctor, patient, measurement agent (e.g., gives asked-for lab tests), and moderator agent (i.e., determines if doctor agent made correct decision) where doctor and patient are allowed to interact in a fixed number of messages. Experiment results are compared against asking an LLM to directly make clinical decisions given full information.

The paper has several strengths. The experiments are extensive: the results cover 7 LLM models,  at least two modalities across text, tabular, and image data, 7 languages, nine specialist settings, and 24 bias evaluations. The paper is well-written and clearly outlines the pipeline of experiments. The figures and tables convey the experiment results well, and the key findings are salient.

The paper has several key weaknesses, which were heavily discussed in the reviews and discussion process. The main concern, which I agree with, is that although the experiment setup is extensive, the clinical practicality of the setting is not well-motivated. The main contribution of this work is the benchmarking of models within this simulated clinical environment. The salient question is then, how realistic is this setup or at least how generalizable are the findings from this environment? If the AgentClinic framework does not translate well to the real-world clinical setting or the results are heavily weighed by extraneous factors like agent mis-specification, then the findings are of little use to the healthcare AI community. By creating a setting where four language agents interact with each other, it is not clear at all if any decline in performance (e.g., Fig 2 and Fig 3) are due to poor performance of any single (or combination of) language models as a patient agent, measurement agent, doctor agent, or moderator agent. While I appreciate Fig 2 exploring the effects of changing which LLM was used for which agent, it is not convincing of the clinical usability --- or generalizability of insights --- in the entire framework.

More detailed ablation or decomposed evaluation experiments are needed. Appendix F covers a few examples of principled analysis to address obvious concerns, several of which were added by reviewer request. However, there are several unanswered questions that make it hard to have confidence in the generalizability of the results.

For example:
 - **(Based on Section F.1 and response to Reviewer mXnj) How forthcoming or accurate is the patient agent?** If performance of AgentClinicQA decreases both when you decrease N from 10 and also when you increase N from 10, wouldn't that suggest that the real problem is how well the doctor agent solicits information and gives diagnosis (especially since the doctor agent sometimes does not give a diagnosis at all)? How would prompt engineering or more tailored system instructions help here? Is it possible that a better directed model would reverse the findings of AgentClinicQA?
 - **(Based on the newly added Section F.3 and response to Reviewer 6dqE) Are results from AgentClinicQA entirely explained by doctor agents not being able to solicit information efficiently?** The new experiment in F.3 annotates the amount of original MedQA information is revealed where GPT-4 was the doctor agent. However, the small size of this experiment (only GPT-4 model was examined) is disappointing. It would be more useful to see if the amount of information revealed explains the differences across ALL discrepancies between MedQA and AgentClinic-MedQA for all LLM models as doctor agents.
 - **How accurate is the moderator agent?** It would appear that if all evaluation is based on yet another language agent, we would want to make sure that the moderator agent is correct.
 - **How accurate is the measurement agent?** In a response to Reviewer 6dqE, the authors claim that this is not of "high relevance to the main takeaways of the paper", especially because there are "agent systems in development ... to retrieve specific patient information in the EHR". I would argue that without critically examining every component of the pipeline, it is difficult to tell what the insights of the results would be. If a database is equally reliable, why not use that? Adding an extra measurement agent here feels unnecessarily complicated, especially when it includes yet another un-ablated component.
 - I will note that the authors include a small human evaluation study (Figure 8) using three individuals with medical degrees. Because the study only evaluates for realism and empathy, I do not consider this helpful for determining performance of each component of AgentClinic.

In making my recommendation, I weighed both the extensive experiment setup with the potentially limited clinical applicability of the setup.

**Additional Comments On Reviewer Discussion:**

The paper had a very active discussion, and I commend both reviewers and authors for being engaged, respectful, and constructive in their responses.

Summary of the main points raised:
 - **Citation of AI Hospital paper [1]**: After a lengthy discussion between Reviewer LHBV who mentioned the similarities between AgentClinic and AI Hospital, the authors agreed to cite AI Hospital and elaborate in the work. I have read through the AI Hospital and agree with Reviewer LHBV's claim that AI Hospital also handles dynamic sequential decision-making (unlike the authors' original claim that AI Hospital did not).
 - **Concerns about the subjective nature of the bias metrics**: Authors acknowledge that the patient agent self-assesses for metrics such as patient compliance, but future work could analyze self-reported compliance and actual adherence for patients. Even after the author responses, I believe that the paper made formalizations of these biases in ways that still appear subjective and not well-explained. The appendix describes the biases explored, but the choice and instantiation of the biases is not grounded in either existing work or clear reasoning. Similarly, the evaluation of the multilingual settings was described as "manually corrected by native speakers" without any detail about how to overcome the complexities of translation or insights that the healthcare AI community would find useful.
 - **Concerns that data leakage may have occurred between MedQA and MIMIC-IV (the two sources of scenarios for AgentClinic)**: Authors acknowledge this is possible, but because MedQA performance across 7 models does not correlate to AgentClinic-MedQA performance, data leakage is unlikely to explain all of the findings. I am personally less convinced by this explanation because data leakage could still be a concern even if the performance is affected by other factors.
 - **AgentClinic is simple compared to clinical settings**: Authors respond that AgentClinic is not meant to replicate real-world settings, but the goal is to highlight challenges that "medical agents" might face including ordering tests and soliciting information. I see this as the main weakness of the work (see above reasoning).
 - **Additional ablation experiments**: Authors added experiments to section F (see above for more details). I believe that these results were insufficient to enable confidence in the results.

[1] https://arxiv.org/pdf/2402.09742

---

> ### Public Comment · ~Liangjun_Zou1 · 2025-09-20
> **1**
>
> 1

---

### Decision · Program_Chairs · 2025-01-22

Reject